# Sleep pressure accumulates in a voltage-gated lipid peroxidation memory

H. Olof Rorsman[1,4], Max A. Müller[2,4], Patrick Z. Liu[1], Laura Garmendia Sanchez[1], Anissa Kempf[1,3], Stefanie Gerbig[2], Bernhard Spengler[2] & Gero Miesenböck[1✉]

Voltage-gated potassium ($K_V$) channels contain cytoplasmically exposed β-subunits[1–5] whose aldo-keto reductase activity[6–8] is required for the homeostatic regulation of sleep[9]. Here we show that Hyperkinetic, the β-subunit of the $K_V1$ channel Shaker in *Drosophila*[7], forms a dynamic lipid peroxidation memory. Information is stored in the oxidation state of Hyperkinetic's nicotinamide adenine dinucleotide phosphate (NADPH) cofactor, which changes when lipid-derived carbonyls[10–13], such as 4-oxo-2-nonenal or an endogenous analogue generated by illuminating a membrane-bound photosensitizer[9,14], abstract an electron pair. $NADP^+$ remains locked in the active site of $K_Vβ$ until membrane depolarization permits its release and replacement with NADPH. Sleep-inducing neurons[15–17] use this voltage-gated oxidoreductase cycle to encode their recent lipid peroxidation history in the collective binary states of their $K_Vβ$ subunits; this biochemical memory influences—and is erased by—spike discharges driving sleep. The presence of a lipid peroxidation sensor at the core of homeostatic sleep control[16,17] suggests that sleep protects neuronal membranes against oxidative damage. Indeed, brain phospholipids are depleted of vulnerable polyunsaturated fatty acyl chains after enforced waking, and slowing the removal of their carbonylic breakdown products increases the demand for sleep.

The pore-forming α-subunits of voltage-gated potassium channels of the $K_V1$ and $K_V4$ families partner with non-membrane-integral β-subunits[1–5] whose sequences exhibit puzzling similarity with aldo-keto reductases[6,7]—enzymes that reduce carbonyls to alcohols via the coupled oxidation of an NADPH cofactor. The isolated β-subunits show weak reductase activity towards a range of model aldehydes in vitro[18,19], relying on NADPH as the electron donor, but whether, on which native carbonyls, and to what end the assembled $K_V$ channel catalyses similar reactions in vivo is unknown. The exceptionally firm grip of $K_Vβ$ on its cofactor[8], which chokes catalysis, deepens the mystery of why an ion channel would be shackled to what appears to be a subpar enzyme.

A hint at a possible answer has come from studies in *Drosophila*, where both the $K_V1$ channel Shaker[20,21] and its β-subunit Hyperkinetic[7] are needed to sustain normal levels of sleep[22,23]. The sleep-regulatory function of the channel complex has been mapped to a small number of sleep-control neurons whose axonal projections target the dorsal fan-shaped body in the central brain[15,17,24] (dFBNs). Sleep need is encoded in the electrical activity of these neurons[16], which fluctuates—in part[24]—because Hyperkinetic modulates the inactivation kinetics of the Shaker current[9]. During waking, electrons leaking from the saturated transport chains of the inner mitochondrial membrane produce superoxide and other reactive oxygen species (ROS), which convert the $K_Vβ$ pool to the $NADP^+$-bound form[9,25]. This prolongs the inactivation time constant of the associated potassium conductance[9,18,26,27], strengthens the repolarizing force that restores the resting membrane potential after each spike, and so enables dFBNs to fire at higher rates[9,28].

Although the source (the mitochondrial electron transport chain) and the receiver (Hyperkinetic in complex with Shaker) of the sleep-promoting redox signal are known[9,25], the mode of communication between mitochondria and potassium channels remains undefined. $K_Vβ$-bound NADPH is an unlikely direct target of ROS, not only because radical-induced hydrogen abstraction (which involves a single electron transfer) will not produce $NADP^+$ (which would require the loss of two electrons). As ROS spread from the inner mitochondrial membrane, they encounter many potential reaction partners before reaching Hyperkinetic at the cell surface. Among the most abundant and vulnerable ROS targets in the immediate vicinity of their site of origin are the polyunsaturated fatty acyl chains (PUFAs) of membrane lipids, whose peroxidation and subsequent fragmentation into carbonyls[10–13] can create chemical functionality fit for the active site of an aldo-keto reductase. In the crystal structure of the mammalian $K_V1.2$–β2 channel complex, the substrate binding pocket is lined with hydrophobic residues and filled with unresolved electron density[4], as would be expected if a diverse group of lipid precursors disintegrated into a heterogeneous mix of apolar ligands. Recombinant $K_Vβ1$ and $K_Vβ2$ reduce synthetic analogues of lipid peroxidation products, such as 4-oxo-2-nonenal (4-ONE), 1-palmitoyl-2-oxovaleroyl-phosphatidylcholine or methylglyoxal, in vitro[18,19], but turnover is so slow that the effect on the concentrations of these molecules in vivo must be minimal. While $K_Vβ$ can therefore have no plausible role in the enzymatic clearance of toxic carbonyls,

[1]Centre for Neural Circuits and Behaviour, University of Oxford, Oxford, UK. [2]Institute of Inorganic and Analytical Chemistry, Justus-Liebig-Universität, Giessen, Germany. [3]Present address: Biozentrum, Universität Basel, Basel, Switzerland. [4]These authors contributed equally: H. Olof Rorsman, Max A. Müller. ✉e-mail: gero.miesenboeck@cncb.ox.ac.uk

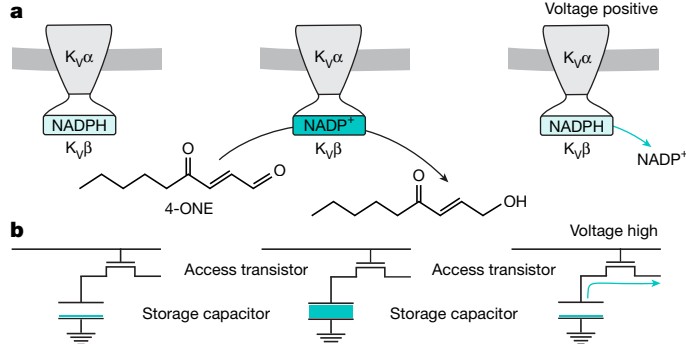

**Fig. 1 | Information storage by $K_V\beta$. a**, The bits 0 (left) and 1 (centre) are stored in the cofactor oxidation state of the $K_V\beta$ subunit. The memory is read out when the membrane potential across $K_V\alpha$ depolarizes and $K_V\beta$ discharges NADP$^+$ (right). **b**, The bits 0 (left) and 1 (centre) are stored in the electrical charge on the capacitor of a DRAM cell. The memory is read out when the voltage across the access transistor gate goes high and the capacitor discharges (right).

the very features that seem detrimental or baroque in a catalyst—the protein's stranglehold on NADP(H) and its linkage to a voltage-gated ion channel—could be essential if the assembly instead functioned as a biochemical memory cell (Fig. 1). Imagine that tight binding of NADP(H) causes the redox reaction to pause at the cofactor-exchange step. Each β-subunit then records a single exposure to an oxidizing substrate by flipping from the NADPH-bound to the NADP$^+$-bound form and stores this bit of information until NADP$^+$ is released and replaced by NADPH (Fig. 1a). The operational logic resembles that of a single-transistor dynamic random-access memory (DRAM) cell[29] (Fig. 1b): $K_V\beta$ corresponds to the storage capacitor of a DRAM cell; the oxidation state of NADP(H) plays the part of the electric charge on the capacitor; and the (low) basal reaction rate is equivalent to the leakage of charge from the capacitor, which gives the memory a finite lifetime that requires periodic refreshment[29]. The analogy would be complete if, akin to the voltage across the transistor that gates access to the storage capacitor in a DRAM chip[29], the membrane potential across the voltage sensors of the α-subunit controlled the rate of cofactor exchange by the β-subunit (Fig. 1).

Here we test several tenets of this model. We examine the lipids of rested and sleep-deprived brains for signs of oxidative damage; measure the effect on sleep of perturbing the clearance of peroxidized lipids; determine whether lipid peroxidation products influence the Shaker current of sleep-control neurons via the active site of Hyperkinetic; and analyse the interplay of voltage sensors and NADP(H) binding sites in the redox regulation of the channel. The results define an autoregulatory loop in which the $K_V1$ channel population encodes the recent lipid peroxidation history of a neuron in the collective binary states of their β-subunits. This biochemical memory (which we equate to the accumulated sleep pressure) is read and erased during subsequent electrical activity, with the action potential frequency set by the fraction of $K_V\beta$ subunits previously loaded with NADP$^+$.

## A lipidomic fingerprint of sleep loss

Because levels of oxidative stress may differ among tissues, brain regions or neuron types[9,30], we collected spatial maps of hundreds of lipids by means of high-resolution scanning microprobe matrix-assisted laser desorption/ionization mass spectrometry imaging (SMALDI-MSI). The lipid maps were acquired by scanning 10-μm-thick cryosections of rested or sleep-deprived brains at a lateral resolution of 5 μm × 5 μm and overlaid on fluorescence images of dFBNs expressing *R23E10-GAL4*-driven[16] mCD8::GFP (Fig. 2a).

Samples within each group had tightly correlated lipid profiles, but differences between groups—that is, between the rested and

sleep-deprived states—were so stark that sleep histories could be accurately inferred from lipid composition alone; a single principal component captured 85% of the overall variance. Fifty-one out of 380 SMALDI-MSI signals annotated as glycerophospholipids and detected exclusively on tissue increased or decreased more than twofold after sleep loss, with a false discovery rate (FDR)-adjusted significance threshold of $P < 0.05$ and little, if any, spatial heterogeneity across the brain (Fig. 2a–c). The identities of 18 of these 51 differentially abundant phospholipids (35%) were confirmed by targeted MS$^2$ fragmentation after HPLC separation of a methyl *tert*-butyl ether extract of brain homogenates (Fig. 2a–c). In many cases these analyses also revealed the detailed fatty acid compositions of the parent species (Fig. 2b,c).

Most lipids with high discriminatory power belonged to one of three classes, which form discernible blocks in the clustergram of Fig. 2b. The glycerophospholipids of rested brains carried inositol, serine, ethanolamine or choline head groups and were enriched in acyl chains with a combined median length of 37.5 carbons and a large degree of unsaturation; the number of double bonds averaged 5.0 ± 2.61 (mean ± s.d.) per lipid, with a median of 5 and a maximum of 12 (Fig. 2b–d). Phospholipids that were present at higher levels in sleep-deprived brains, by contrast, contained mostly choline and ethanolamine head groups, shorter acyl chains with a combined median length of 33.5 carbons, and many fewer double bonds than those in rested flies; the number of double bonds averaged 2.0 ± 2.03 (mean ± s.d.) per lipid, with a median of 2 (Fig. 2b–d). The third distinctive lipid class consisted of several species of phosphatidic acid, whose levels declined after sleep deprivation (Fig. 2b–d). Phosphatidic acid occupies a central position in the biosynthetic pathways of all glycerophospholipids[31,32] and promotes mitochondrial fusion when generated locally by a dedicated phospholipase D (mitoPLD)[33]. Impaired mitoPLD activity in dFBNs causes sleep loss[25].

The lipidomic fingerprint of sleep-deprived brains indicates that their membranes are depleted of PUFAs, presumably as a consequence of oxidative damage, leaving behind a greater proportion of largely saturated phospholipids (Fig. 2d). The picture during rest is consistent with membrane repair via glycerophospholipid biosynthesis from phosphatidic acid precursors[31,32] and a reversal of the mitochondrial fragmentation that commonly accompanies periods of oxidative stress[34], including sleep deprivation[25].

## Lipid-derived carbonyls promote sleep

The peroxidation of membrane lipids begins[11,12] with the abstraction of a *bis*-allylic hydrogen from a PUFA chain by a radical oxidant such as HOO• (the conjugate acid of $O_2^-$) or •OH. The resulting lipid radical reacts with $O_2$ to form a lipid peroxyl radical, which propagates the chain by abstracting a hydrogen from another PUFA, generating a new lipid radical and a lipid hydroperoxide[10–13]. The reaction continues until two radicals combine in a termination step. The lipid hydroperoxides produced along the way undergo a series of rearrangements and scissions that give rise to a variety of short- and medium-chain carbonyl breakdown products[10–13], including the potential $K_V\beta$ substrate[18,19] 4-ONE.

Operating behind a primary bastion of enzymatic and non-enzymatic antioxidants[35], soluble short-chain dehydrogenases/reductases, such as carbonyl reductase 1 in mammals[36,37] and its functional homologue sniffer in *Drosophila*[38,39], form a second defensive ring against lipid peroxidation-derived carbonyls. We examined whether breaching and mending these secondary defences would recapitulate the well-documented effects on sleep of pro- and antioxidant manipulations[9,30,40]. Indeed, hemizygous male carriers of the X-linked hypomorphic *sniffer* allele *sni$^1$* showed increased sleep durations during the day and night (Fig. 3a–c and Extended Data Fig. 1a), owing to vastly extended, hyperconsolidated sleep episodes (Extended Data Fig. 1b,c), at an age before widespread neurodegeneration[38] produced locomotor deficits that could have been mistaken for sleep (Extended Data

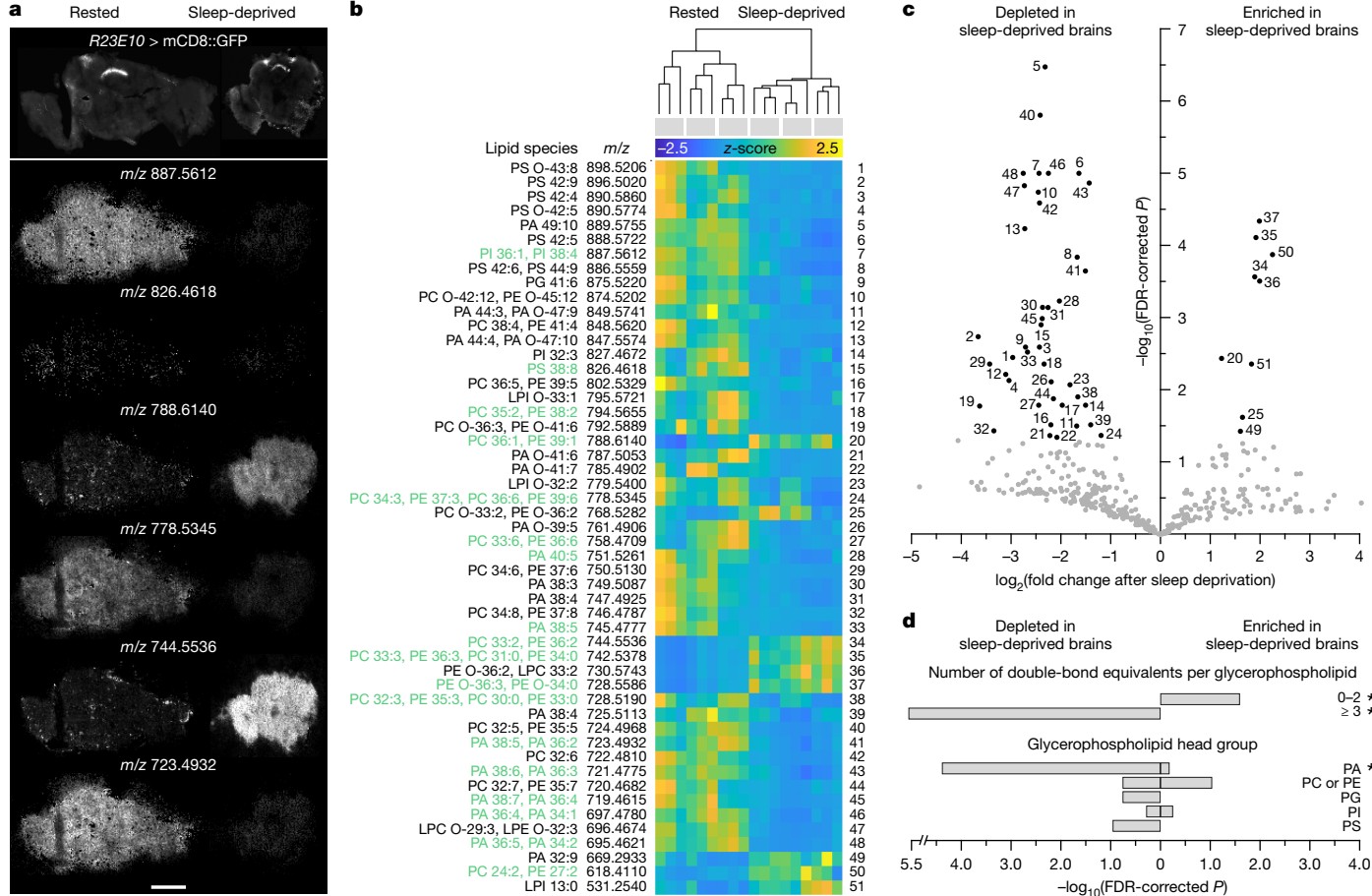

**Fig. 2 | Sleep deprivation depletes brain phospholipids of polyunsaturated fatty acids. a**, Example fluorescence (top) and positive-ion SMALDI-MS images (bottom) of cryosections containing dFBNs marked with mCD8::GFP. The sections were cut from rested (left) or sleep-deprived brains (right). SMALDI-MS images show, from top to bottom, the spatial distributions of phosphatidylinositol 18:2/20:2 (*m/z* 887.5612, [M+Na]⁺), phosphatidylserine 18:3/20:5 (*m/z* 826.4618, [M+Na]⁺), phosphatidylcholine 18:0/18:1 (*m/z* 788.6140, [M + H]⁺), phosphatidylcholine 18:3/18:3 (*m/z* 778.5345, [M + H]⁺), phosphatidylethanolamine 18:1/18:1 (*m/z* 744.5536, [M + H]⁺) and phosphatidic acid 18:2/20:3 (*m/z* 723.4932, [M + H]⁺). Scale bar, 200 μm. **b**, Hierarchical clustering of rested and sleep-deprived brains according to their glycerophospholipid profiles. Heat maps show the *z*-scored intensities of *m/z* signals differing with sleep history at an FDR-adjusted *P* < 0.05 (two-sided *t*-test). Lipids detected in MS² fragmentation experiments are annotated in green in the list of molecular assignments on the left. Each column represents a different cryosection (*n* = 9 per condition); sections of the same brain (*n* = 3 per condition) are grouped by grey bars on top. **c**, Volcano plot of sleep history-dependent changes in 380 *m/z* signals annotated as glycerophospholipids. Signals with more than twofold intensity changes and FDR-corrected *P* < 0.05 (two-sided *t*-test) are indicated in black. Numerical labels reference data points to lipid annotations in **b**. **d**, Features overrepresented in the subset of 51 differentially abundant lipids against the background set of all 380 glycerophospholipids. Asterisks indicate significant enrichment scores (FDR-corrected *P* < 0.05, Fisher's exact test). Because phosphatidylcholine and phosphatidylethanolamine lipids cannot be distinguished by exact mass alone, they are grouped as a single feature. LPC, lysophosphatidylcholine; LPE, lysophosphatidylethanolamine; LPI, lysophosphatidylinositol; PA, phosphatidic acid; PC, phosphatidylcholine; PE, phosphatidylethanolamine; PG, phosphatidylglycerol; PI, phosphatidylinositol; PS, phosphatidylserine; O-, alkyl ether linkage.

Fig. 1d). Sleep returned to or below wild-type levels when *sni¹* mutants expressed a *UAS-sni* rescue transgene³⁸ (Fig. 3c and Extended Data Fig. 1a), and similarly when the alternative oxidase AOX, which shunts surplus electrons from ubiquinone to H₂O, capped mitochondrial ROS production⁹,⁴¹ (Fig. 3a,c), or when the putative carbonyl sensor Hyperkinetic was removed by RNA-mediated interference (RNAi), either pan-neuronally or in dFBNs of *sni¹* mutant flies (Fig. 3b,c). These data place lipid peroxidation products downstream of mitochondrial respiration in the signalling chain that terminates on the Hyperkinetic pool of dFBNs to raise the pressure to sleep⁹.

## A redox memory of lipid peroxidation

To determine whether lipid peroxidation-derived carbonyls could alter the oxidation state of Hyperkinetic's cofactor, we obtained whole-cell voltage-clamp recordings from dFBNs and estimated the NADP⁺:NADPH ratio of the K$_V$β population from the bi-exponential inactivation kinetics of the A-type current ($I_A$) (Extended Data Fig. 2): a reduced cofactor increases, whereas an oxidized cofactor decreases, the rate of channel inactivation⁹,¹⁸,²⁶,²⁷. If PUFA-derived carbonyls are endogenous electron acceptors at the active site of K$_V$β, their ballooning levels in *sni¹* mutants³⁸,³⁹ should drive the Shaker–Hyperkinetic complex into the NADP⁺-bound, slowly inactivating state. Increases in the fast and slow inactivation time constants ($\tau_{fast}$ and $\tau_{slow}$, respectively) of the A-type current relative to wild-type flies indicate that this was indeed the case (Fig. 4a,b).

Plasma membrane-anchored miniSOG¹⁴ allowed us to switch the cofactor acutely to the oxidized state⁹ and follow its fate thereafter. The exposed chromophore of this light-oxygen-voltage-sensing (LOV) domain protein¹⁴ transfers the energy of blue light efficiently to O₂, producing singlet oxygen (¹O₂) which—presumably indirectly, via a burst of lipid peroxidation—converts the channel population

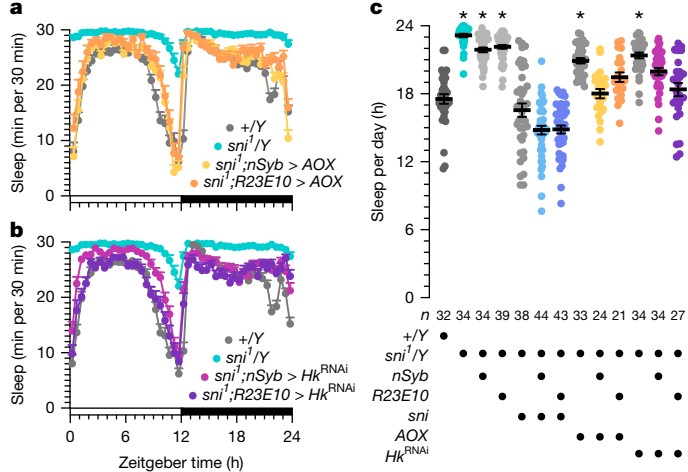

**Fig. 3 | Lipid peroxidation products are intermediates in the signalling chain that couples mitochondrial electron transport to sleep. a**, The *nSyb-GAL4-* or *R23E10-GAL4*-driven expression of AOX in hemizygous *sni[1]* mutant males fully or partially restores wild-type sleep (two-way repeated-measures ANOVA with Holm−Šídák test; sample sizes in **c**). The sleep profiles of *sni[1]* mutants with pan-neuronal expression of AOX differ from those of *sni[1]* mutants ($P < 0.0001$) but not of wild-type flies ($P = 0.0589$), whereas the sleep profiles of *sni[1]* mutants with dFBN expression of AOX differ from those of both *sni[1]* mutants ($P < 0.0001$) and wild-type flies ($P = 0.0007$). **b**, *nSyb-GAL4-* or *R23E10-GAL4*-restricted interference with the expression of Hyperkinetic in hemizygous *sni[1]* mutant males partially or fully restores wild-type sleep (two-way repeated-measures ANOVA with Holm−Šídák test; sample sizes in **c**). The sleep profiles of *sni[1]* mutants with pan-neuronal expression of $Hk^{RNAi}$ differ from those of both *sni[1]* mutants ($P < 0.0001$) and wild-type flies ($P < 0.0001$), whereas the sleep profiles of *sni[1]* mutants with dFBN expression of $Hk^{RNAi}$ differ from those of *sni[1]* mutants ($P < 0.0001$) but not of wild-type flies ($P = 0.1344$). **c**, Sleep in hemizygous males carrying the *sni[1]* allele differs from wild-type ($P < 0.0001$; Kruskal–Wallis ANOVA with Dunn's test) but returns to or below control level if carriers also express sniffer (sni), AOX or $Hk^{RNAi}$ pan-neuronally under the control of *nSyb-GAL4* (sni: $P = 0.1128$; AOX: $P > 0.9999$; $Hk^{RNAi}$: $P = 0.0601$) or in dFBNs under the control of *R23E10-GAL4* (sni: $P = 0.1151$; AOX: $P = 0.6694$; $Hk^{RNAi}$: $P > 0.9999$). Note that the expression of the *UAS-sni* transgene appears leaky, as the sleep phenotype of *sni[1]* mutants is rescued in the absence of a *GAL4* driver ($P > 0.9999$). Data are mean ± s.e.m.; *n*, number of flies; asterisks indicate significant differences ($P < 0.05$) from wild type in planned pairwise comparisons. For statistical details see Supplementary Table 1.

to the NADP$^+$-bound form and induces sleep[9]. The oxidation of the cofactor was detected as an increase in the fast and slow inactivation time constants after 9 min of blue light exposure, from initial mean values of 5.8 and 35 ms to final averages of 8.2 and 59 ms (Fig. 4c,d). When the membrane potential was clamped at −80 mV, $\tau_{fast}$ and $\tau_{slow}$ stayed stably elevated for 20 min after the light-driven $^1O_2$ generation stopped (Fig. 4c,d), consistent with a negligible rate of spontaneous NADP$^+$ exchange[8,18,27] that allows Hyperkinetic to retain a memory of an earlier encounter with an oxidizing substrate, even if that molecule is itself short-lived (estimated intracellular half-life[10] of lipid-derived carbonyls <4 s).

In a direct test of the idea that PUFA-derived carbonyls are prominent among these substrates, we filled dFBNs through the patch pipette with the synthetic lipid peroxidation products 4-ONE or 4-hydroxynonenal (4-HNE)[10,12,13]. Owing to their inherent reactivity and membrane-permeability, the equilibration of these carbonyls within the neuronal arbor was governed by complex reaction–diffusion kinetics that made their concentration profiles difficult to predict[13] and, in all likelihood, neither spatially uniform nor temporally stationary during the course of a recording. 4-ONE and 4-HNE are estimated (with large uncertainty) to be present in cells in the low to sub-micromolar range under basal conditions but reach millimolar concentrations

during periods of oxidative stress[10]. Although 4-HNE is viewed as a useful marker of lipid peroxidation because monoclonal antibodies can detect its protein adducts[42], mammalian $K_V\beta2$ in vitro shows detectable catalytic activity only towards 4-ONE[19]. If the substrate preferences of *Drosophila* Hyperkinetic were similar, 4-HNE could serve as an ideal control to distinguish effects due to the enzymatic conversion of reactive carbonyls from those potentially caused by indiscriminate protein modification[13].

Comparisons of $I_A$ inactivation kinetics immediately after break-in and 10 min later revealed a clear slowing of the fast and slow time constants, with effect sizes similar to those after the miniSOG-driven photogeneration of ROS (Fig. 4e,f) or a night of mechanical sleep deprivation (Fig. 4g). Changes were seen only in dFBNs perfused with 50 μM 4-ONE; 200 μM 4-HNE, the addition of 0.15% methyl acetate vehicle to the intracellular solution, or the passage of time alone had no effect (Fig. 4g and Extended Data Fig. 3a). When the cells were held at −80 mV in 4-ONE for extended periods, the inactivation time constants completed much of their climbs to higher plateaus within the first 10 min and remained there for the rest of the recordings (Fig. 4f). Because each neuron in this experimental configuration was connected to a practically infinite reservoir of 4-ONE, however, the persistent slowing of inactivation could reflect continuous turnover of substrate rather than a lasting switch in the oxidation state of the cofactor; it can therefore not speak as unequivocally to the longevity of the redox memory as the enduring increase of $\tau_{fast}$ and $\tau_{slow}$ in miniSOG-expressing dFBNs after a finite light exposure can (Fig. 4d).

Membrane resistances, membrane time constants, and the amplitude of the non-A-type potassium current remained approximately constant over the course of 30 min, but the magnitude of $I_A$ slowly declined (Extended Data Figs. 3a,b and 4). This trend is likely to reflect closed-state inactivation[43] rather than a gradual loss of voltage control over a portion of the channels before an increase in access resistance would have prompted us to terminate the recording: series resistances stayed within stable limits for 30 min, irrespective of the presence of 4-ONE or changes in command potential or the inactivation kinetics of $I_A$ (Extended Data Fig. 3c), but the steady-state half-inactivation voltages drifted towards more hyperpolarized potentials[43] (Extended Data Fig. 3d). Because the same slow rundown of $I_A$ was also observed in the absence of 4-ONE (Extended Data Fig. 3a), after miniSOG stimulation (Extended Data Fig. 4b), and in homozygous *Hyperkinetic*-null mutants (below), the effect cannot be explained by a direct irreversible 4-ONE hit on the β-subunit.

For the most stringent proof that 4-ONE altered the Shaker current via its reduction at the active site of $K_V\beta$ (as opposed to an off-target modification on the channel or elsewhere), we expressed transgenes encoding catalytically active or dead Hyperkinetic[44] under *R23E10-GAL4* control in dFBNs of *Hyperkinetic*-null mutant ($Hk^1/Hk^1$) flies[9]. Infiltrating the Shaker channel with a β-subunit devoid of oxidoreductase activity[18,27,44] (Hk(K289M)) rendered the fast and slow components of A-type inactivation resistant to 4-ONE, whereas the incorporation of functional $K_V\beta$ preserved the sensitivity of the channel (Fig. 4g and Extended Data Fig. 5).

Impaired carbonyl clearance, the photogeneration of ROS, and synthetic 4-ONE exerted indistinguishable effects on $I_A$ in voltage-clamp recordings (Fig. 4a–f), but only carriage of the *sni[1]* mutation or miniSOG-mediated photooxidation also enhanced the spiking response of dFBNs to membrane depolarization (Fig. 5a,b). The delivery of 4-ONE through a patch electrode at the soma did not (Fig. 5c), in all likelihood because the diffusion time of 4-ONE to dFBN axons, which appear rich in Hyperkinetic but are connected to the cell body through a long, thin primary neurite (Fig. 5d), exceeded the brief intracellular half-life of the molecule[10,13]. The release of endogenous lipid peroxidation products, by contrast, whether instigated by miniSOG or amplified by a lack of sniffer, was sufficiently decentralized to be felt also in remote parts of the neuron. The variable spread of externally supplied and internally

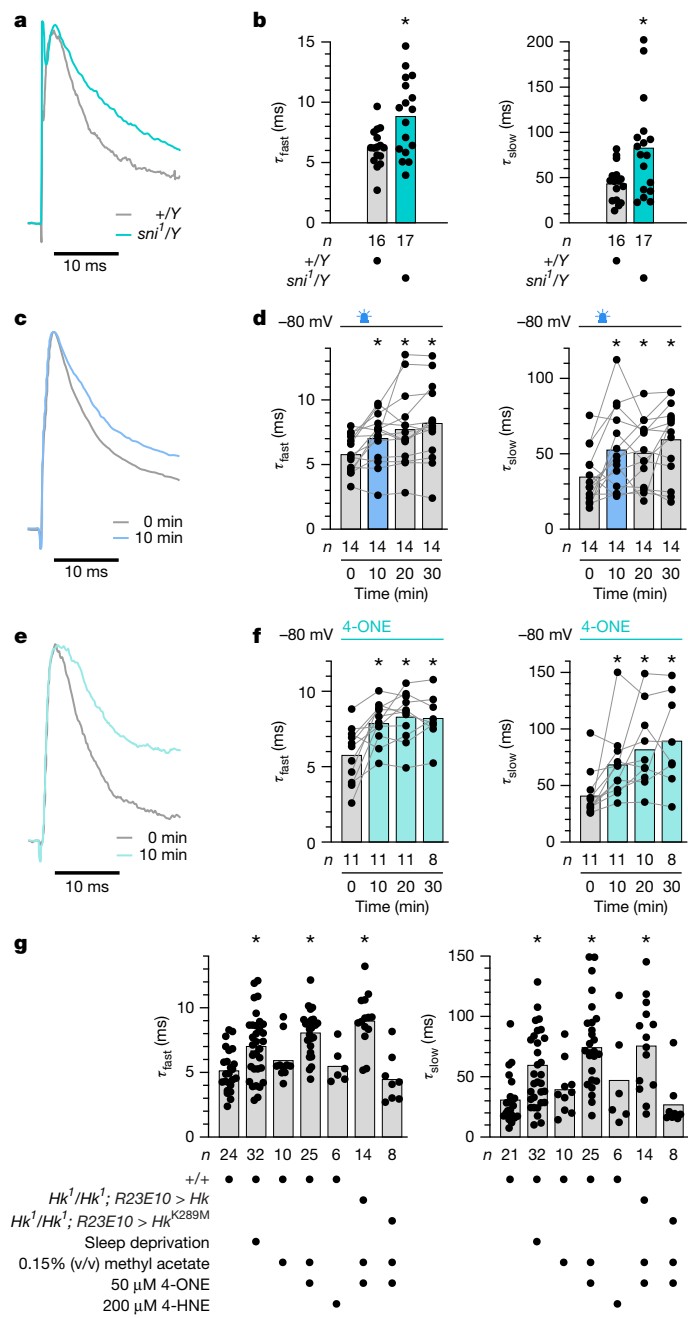

**Fig. 4 | Lipid peroxidation products alter the inactivation kinetics of $I_A$ via the active site of $K_V\beta$. a,b**, The $sni^1$ allele increases the fast and slow inactivation time constants of $I_A$ in dFBNs of hemizygous carriers (turquoise) relative to wild-type males (grey) (**b**; $\tau_{fast}$: $P = 0.0060$, two-sided $t$-test; $\tau_{slow}$: $P = 0.0253$, two-sided Mann–Whitney test; examples of peak-normalized $I_A$ evoked by voltage steps to +30 mV in **a**). **c,d**, dFBNs expressing miniSOG were held at –80 mV, except during the voltage protocols required to measure $I_A$. A 9-min exposure to blue light between the 0- and 10-min time points (**d**; blue) increases the fast and slow inactivation time constants of $I_A$ above their pre-illumination baselines (**d**; $\tau_{fast}$: $P = 0.0133$; $\tau_{slow}$: $P = 0.0041$; repeated-measures ANOVA; examples of peak-normalized $I_A$ evoked in the same dFBN by voltage steps to +30 mV in **c**). **e,f**, dFBNs were held at –80 mV, except during the voltage protocols required to measure $I_A$. The inclusion of 50 μM 4-ONE in the intracellular solution (**f**) increases the fast and slow inactivation time constants of $I_A$ above the baselines recorded immediately after break-in (**f**; $\tau_{fast}$: $P = 0.0015$; $\tau_{slow}$: $P = 0.0010$; mixed-effects model; examples of peak-normalized $I_A$ evoked in the same dFBN by voltage steps to +30 mV in **e**). **g**, dFBNs were held at –80 mV, except during the voltage protocols required to measure $I_A$. At 10 min after break-in, the inclusion of 50 μM 4-ONE, but not of 200 μM 4-HNE, in the intracellular solution increases the fast and slow inactivation time constants of $I_A$ from control to sleep-deprived levels, provided dFBNs express catalytically competent Hyperkinetic ($\tau_{fast}$: $P < 0.0001$; $\tau_{slow}$: $P < 0.0001$; Kruskal–Wallis ANOVA). Columns show population averages; dots represent individual cells; $n$, number of cells; asterisks indicate significant differences ($P < 0.05$) relative to the 0-min time point or control levels in planned pairwise comparisons by Holm–Šídák or Dunn's test. For statistical details see Supplementary Table 1.

in the cytoplasm exceeds that of NADP[+] by at least 40-fold[46]. Such a mechanism would confirm a long-suspected quirk in the enzymatic cycle of $K_V\beta$ and offer a rationale for the association of the protein with a voltage-gated ion channel[8,27].

We tested the prediction that cofactor exchange is voltage-controlled in both of our experimental configurations, using either the photogeneration of ROS by miniSOG (Fig. 6a,b and Extended Data Fig. 6a) or the inclusion of 50 μM 4-ONE in the intracellular solution (Fig. 6c,d and Extended Data Fig. 6b) to load $K_V\beta$ with NADP[+]. Following the expected increases of the fast and slow inactivation time constants at 10 min after break-in, dFBNs were taken through simulated 20-min spike trains at 10 Hz under voltage clamp, with each 'action potential' consisting of a 3-ms somatic depolarization to +10 mV. Measurements of $\tau_{fast}$ and $\tau_{slow}$ after this sequence of voltage steps (that is, at 30 min after break-in) showed full reversals of the initial increases driven by miniSOG or 4-ONE (Fig. 6a–d). These reversals were themselves reversible: when dFBNs filled with 4-ONE were held at –80 mV for a further 10 min, the large surplus of 4-ONE in the patch pipette once again drove increases in both inactivation time constants (Fig. 6d), whereas a second 9-min light exposure accomplished the same for miniSOG-expressing cells (Fig. 6b).

Occasionally, the reversal protocol pushed the inactivation time constants below their original baselines, suggesting that depolarization dissipated not only the oxidative strain applied by 4-ONE or mini-SOG but also the internally sourced pressure already integrated by the channel complex before the experiment began. Consistent with this idea, dFBNs expressing catalytically inactive[18,27,44] Hk(K289M), which cannot form a redox memory (Fig. 4g and Extended Data Fig. 5a,b), often exhibit the fastest-inactivating A-type currents at baseline[9] and no modulation by 4-ONE or subsequent voltage changes (Extended Data Fig. 5a,b).

The ability to remember exposures to lipid peroxidation products is an intrinsic property of $K_V1$ channels, shared by neurons other than dFBNs (Extended Data Fig. 7a–e) and present in mammals, with broad– although not limitless[19]–carbonyl selectivity. When HEK-293 cells coexpressing mouse $K_V1.4$ and $K_V\beta2$ were incubated in extracellular medium containing 12 mM methylglyoxal, a membrane-permeable

generated carbonyls will matter little in measurements of voltage-gated potassium currents, which for space-clamp reasons are dominated by channels near the somatic recording site[45] (Fig. 5d–f), but come to the fore in recordings of action potentials if the spike initiation zone lies outside the diffusion distance of 4-ONE.

## Voltage changes clear the redox memory

The stability of cofactor binding suggests that each conversion of $K_V\beta$ to the NADP[+]-bound state leaves an imprint lasting many minutes (Fig. 4c,d). We equate this imprint–or, more accurately, the imprint on the oxidation state of the Hyperkinetic pool of a dFBN as a whole–with a log of accumulated sleep pressure (Fig. 4g). As in a digital recording, the binary states of many elementary memory cells thus quantize a continuous variable, with a resolution determined by the number of single-bit units. Because sleep pressure is discharged via the electrical activity of dFBNs[16,17], action potentials should erase this memory by releasing NADP[+] and allowing its replacement with NADPH, whose concentration

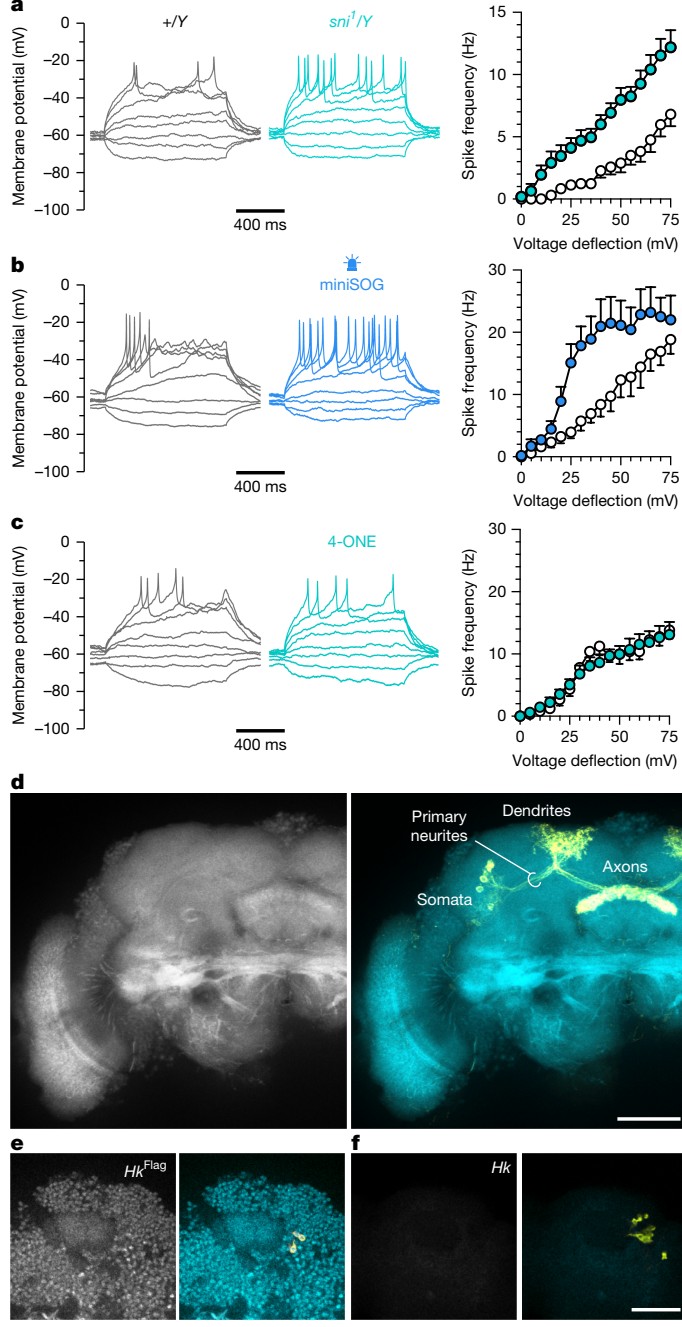

**Fig. 5 | Lipid peroxidation products increase the excitability of dFBNs via axonal $K_V\beta$. a–c**, Example voltage responses to current steps (left) and voltage-spike frequency functions (right; mean ± s.e.m.) of dFBNs. In each neuron, the size of the unitary current step was adjusted to produce a 5-mV deflection from a resting potential of −60 ± 5 mV. The *sni[1]* mutation steepens the voltage-spike frequency function of hemizygous carriers (turquoise, *n* = 11 cells) relative to wild-type males (grey, *n* = 10 cells) (**a**; genotype effect: *P* = 0.0003; current × genotype interaction: *P* < 0.0001; two-way repeated-measures ANOVA). Blue illumination for 9 min steepens the voltage-spike frequency function of dFBNs expressing miniSOG (blue, *n* = 6 cells) relative to controls kept in darkness (grey, *n* = 7 cells) (**b**; illumination effect: *P* = 0.0235; current × illumination interaction: *P* = 0.0008; two-way repeated-measures ANOVA). The inclusion of 50 µM 4-ONE in the intracellular solution (turquoise, *n* = 12 cells) does not steepen the voltage-spike frequency function relative to controls at the 10-min time point (grey, *n* = 10 cells) (**c**; 4-ONE effect: *P* = 0.9052; current × 4-ONE interaction: *P* = 0.7846; two-way repeated-measures ANOVA). **d–f**, Summed intensity projection of a stack of 22 confocal image planes (axial spacing 0.7973 µm) through the fan-shaped body of a fly carrying the *Hk*[Flag] allele (**d**) and single confocal image planes through the somatic regions of flies carrying the *Hk*[Flag] allele (**e**) or an unmodified *Hk* locus (**f**). Specimens were stained with anti-Flag antibody (left); native *R23E10-GAL4*-driven mCD8::GFP fluorescence (yellow) is overlaid on the anti-Flag channel (turquoise) on the right. Scale bars, 50 µm. For statistical details see Supplementary Table 1.

dicarbonyl that serves as an established substrate[19] for $K_V\beta2$, the fast and slow inactivation time constants of the reconstituted A-type current rose and remained durably elevated for 20 min after the removal of methylglyoxal (Extended Data Fig. 7f–h). As in dFBNs, the memory of the carbonyl exposure was retained if the membrane containing the $K_V1.4$–β2 complex was clamped at −80 mV but forgotten during a simulated 20-min spike train at 10 Hz (Extended Data Fig. 7i,j).

## Discussion

Our experiments suggest that $K_V\beta$ subunits are voltage-gated memories used by neurons and other excitable cells to keep score of lipid peroxidation events. Information is stored in the oxidation state of a nicotinamide molecule bound so tightly that it should perhaps be considered a prosthetic group rather than a cofactor, even though two steps in a stop-and-go redox reaction cycle—hydride transfer and nicotinamide exchange—are used to move data to and from memory.

Definitive proof that peroxidized lipids or their breakdown products are endogenous $K_V\beta$ substrates would require their co-purification with the native ion channel—a formidable challenge not only because of the expected molecular heterogeneity of these substrates[10–13], but also because their binding to $K_V\beta$ may be much looser than that of NADP(H); in contrast to the nucleotide binding cleft, which resembles a locked vice, the active site appears wide open in the crystal structure[8].

Our experiments also suggest, but do not prove beyond doubt, that sleep loss causes widespread lipid peroxidation in the brain. Definitive proof would require a demonstration that peroxidation products accumulate, rather than that polyunsaturated phospholipids are depleted, as we have shown. Most previous attempts to measure lipid peroxidation after sleep loss have focused on a single end product, malondialdehyde[10], and yielded variable results[47–50], perhaps because the picture seen through the lens of malondialdehyde is incomplete[42] or because the assays used for its detection report tissue oxidizability during analysis rather than pre-existing levels of peroxidized lipids. Our own attempts to quantify endogenous 4-ONE after sleep deprivation were thwarted by the short half-life[10,13] of the molecule in tissue: while SMALDI-MSI could easily detect 100 µM 4-ONE in isolation, the signal vanished when the same quantity of standard was spiked onto a brain section full of endogenous carbonyl-reactive nucleophiles[10,13] and enzymes[38] (Extended Data Fig. 8a). Trace amounts of 4-ONE captured by Girard's reagent during the derivatization of rested, but not sleep-deprived, brains must reflect the oxidation of the undepleted PUFA pools of these samples in vitro because the *sni[1]* mutation, which would have raised 4-ONE levels in vivo[38,39], caused no discernible increase at the time of measurement (Extended Data Fig. 8b).

While our interpretation of sleep pressure as mitochondrially determined[9,25] lipid peroxidation history demands that sleep-control neurons are equipped to sense and respond to this history, integral redox sensors are a general feature of $K_V1$ (and also some $K_V4$) channels[1–5] in virtually all neurons and many other electrically excitable cells. What could be the purpose of β-subunits in this wider context? Redox control of electrical activity may protect non-renewable cells with high respiratory capacity and extensive membrane systems—such as those of the brain and heart—from oxidative damage if the electron supply to their mitochondria surpasses the demands of ATP synthesis[25]. Depending on where this relief valve opens, the consequences may range from a few extraneous action potentials[28] (in order to re-balance energy consumption with mitochondrial electron flux) to the induction of sleep[9].

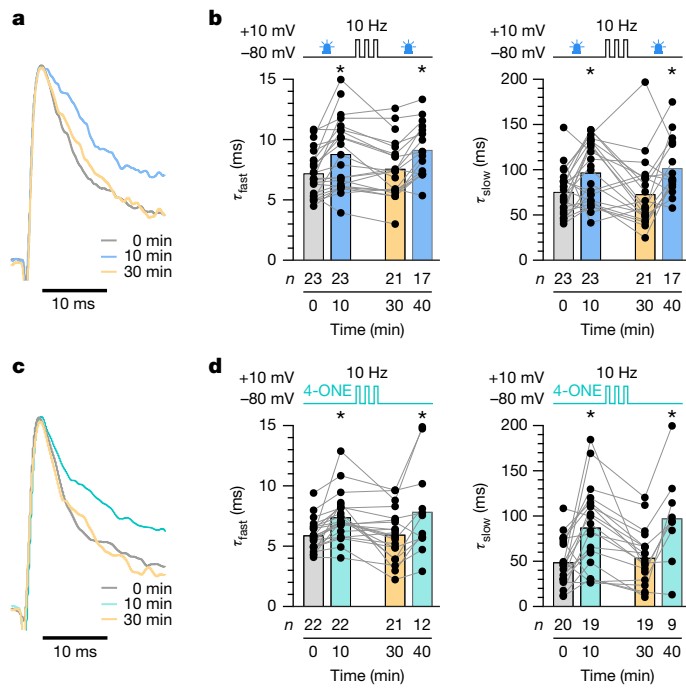

**Fig. 6 | Membrane depolarization clears the lipid peroxidation memory.**
**a**,**b**, dFBNs expressing miniSOG were held at −80 mV in the intervals of 0–10 and 30–40 min (except during the voltage protocols required to measure $I_A$) and repeatedly step-depolarized to +10 mV (3 ms, 10 Hz) between 10 and 30 min. Nine-minute exposures to blue light (between the 0- and 10-min and the 30- and 40-min time points) increase the fast and slow inactivation time constants of $I_A$ above their pre-illumination baselines (**b**; blue versus grey shading); a series of depolarization steps between 10 and 30 min reverses this increase (**b**; yellow shading; $\tau_{fast}$: $P < 0.0001$; $\tau_{slow}$: $P = 0.0008$; mixed-effects model; examples of peak-normalized $I_A$ evoked in the same dFBN by voltage steps to +30 mV in **a**). **c**,**d**, dFBNs were held at −80 mV in the intervals of 0–10 and 30–40 min (except during the voltage protocols required to measure $I_A$) and repeatedly step-depolarized to +10 mV (3 ms, 10 Hz) between 10 and 30 min. The inclusion of 50 µM 4-ONE in the intracellular solution increases the fast and slow inactivation time constants of $I_A$ above the baselines recorded immediately after break-in (**d**; turquoise versus grey shading); a series of depolarization steps between 10 and 30 min counteracts this increase despite the continuous presence of 4-ONE (**d**; yellow shading; $\tau_{fast}$: $P = 0.0053$; $\tau_{slow}$: $P = 0.0012$; mixed-effects model; examples of peak-normalized $I_A$ evoked in the same dFBN by voltage steps to +30 mV in **c**). Columns show population averages; dots represent individual cells; $n$, number of cells; asterisks indicate significant differences ($P < 0.05$) relative to the 0-min time point in planned pairwise comparisons by Holm–Šídák test. For statistical details see Supplementary Table 1.

Just as sodium spikes are universal information carriers filled with distinctive meaning by the different neurons that emit them, excitability control by $K_V\beta$ may be a general mechanism co-opted by dFBNs for the special purpose of regulating sleep.

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

## Methods

### *Drosophila* strains and culture

Flies were reared on media of cornmeal (62.5 g l⁻¹), inactive yeast powder (25 g l⁻¹), agar (6.75 g l⁻¹), molasses (37.5 ml l⁻¹), propionic acid (4.2 ml l⁻¹), tegosept (1.4 g l⁻¹) and ethanol (7 ml l⁻¹) on a 12 h light:12 h dark cycle at 25 °C. All electrophysiological and lipidomic analyses (with the exception of studies of the effects of the *sni¹* mutation) were performed on randomly selected female flies aged 2–6 days post eclosion. Experimental flies were heterozygous for all transgenes and homozygous for either a wild-type or mutant (*Hk¹*) *Hyperkinetic* allele[51,52], as stated. The *R23E10-GAL4* driver[16,53] controlled the expression of the fluorescent label mCD8::GFP in dFBNs, along with an N-myristoylated covalent hexamer (myr-MS6T2) of the singlet oxygen generator miniSOG[54] or catalytically defective (Hk(K289M)) or functional versions of Hyperkinetic[44], as indicated. The *Dh31-GAL4* line[55] targeted mCD8::GFP to neurons of the pars intercerebralis.

Because *sni* is X-linked[38], it was most expedient to investigate its function in males. In behavioural experiments or 4-ONE analyses, hemizygous carriers of the *sni¹* allele coexpressed *UAS-sni*[38], *UAS-AOX*[56] or *UAS-Hk*^RNAi (47805GD)[57] transgenes, either pan-neuronally[58] under the control of *nSyb-GAL4* or in dFBNs[16,53] under the control of *R23E10-GAL4*, as noted. For electrophysiological recordings, dFBNs of hemizygous *sni¹* mutants and wild-type males were marked with *R23E10-GAL4*-driven mCD8::GFP.

A *Hyperkinetic* allele encoding an in-frame fusion to an N-terminal Flag epitope (*Hk*^Flag) was created through homology-dependent repair of a CRISPR–Cas9-generated double-strand break (WellGenetics). The Flag tag was inserted immediately after the initiating methionine of isoforms Hk-PK, Hk-PE, Hk-PL, and Hk-PM and connected to the remainder of the protein via a flexible linker (4× Gly-Gly-Ser).

### Sleep measurements and sleep deprivation

Females or hemizygous *sni¹* mutant males[38] aged 2–5 days were individually inserted into 65-mm glass tubes, loaded into *Drosophila* Activity Monitors (Trikinetics), and housed under 12 h light:12 h dark conditions. Flies were allowed to adapt to the monitors for a day, and the activity counts during the following two 24-h periods were averaged. Inactivity periods of >5 min were classified as sleep[59,60] (Sleep and Circadian Analysis MATLAB program[61]). Immobile flies (<2 beam breaks per 24 h) were manually excluded.

To deprive flies of sleep, a spring-loaded platform stacked with Trikinetics monitors was slowly tilted by an electric motor, released, and allowed to snap back to its original position[62]. The mechanical cycles lasted 10 s and were repeated continuously for 12 h, beginning at zeitgeber time 12.

### SMALDI mass spectrometry imaging

Dissected brains of rested and sleep-deprived flies were placed on PTFE-printed glass slides (Electron Microscopy Sciences), covered with ~3–5 µl gelatin (5% w/v in water), and snap frozen for shipping. For sectioning, dissected brains were thawed, suspended in 20 µl 5% gelatin, and transferred to a gelatin plateau created by removing the top half of a frozen block of 5% gelatin in a cryostat (Microm HM 525, ThermoFisher). After allowing the samples to refreeze during 10 min in the cryostat chamber, 10-µm sections were cut and thaw-mounted onto glass slides. The sections were imaged in fluorescence (BX41, Olympus) and reflected light mode (VHX 5000, Keyence) and stored at −80 °C until further use.

For SMALDI-MSI[63], the brain sections were thawed in a desiccator and spray-coated with 80 µl of a freshly prepared solution of 2,5-dihydroxybenzoic acid (DHB, Merck) using a SMALDIPrep ultrafine pneumatic spraying system (TransMIT GmbH). The DHB solution contained 60 mg of DHB in 999 µl acetone, 999 µl water, and 2 µl pure trifluoroacetic acid (TFA, Merck). In samples destined for 4-ONE analysis, a chemical derivatization step with Girard's reagent T (GirT, TCI Chemicals) preceded the application of the DHB matrix[64]. The samples were spray-coated with 35 µl of a freshly prepared solution of 15 mg ml⁻¹ GirT in a 7:3 mixture of methanol and water containing 0.2% (v/v) TFA and incubated in a desiccator at room temperature for 2 h. Standards were prepared by applying 5-µl droplets of a tenfold dilution series of 4-ONE (Cayman Chemical) in methyl acetate, from 100 µM to 10 nM, onto blank glass slides or slides containing brain sections of rested flies. Standards underwent the same GirT-derivatization and matrix application steps as analytical samples.

A home-built SMALDI-MS imaging ion source based on an AP-SMALDI5 AF system (TransMIT GmbH) was coupled to an orbital trapping mass spectrometer (Q Exactive, ThermoFisher). Mass spectra were acquired at a mass resolution of 140,000 in positive-ion mode. A high voltage of 4 kV was applied to the sample holder. The standard pixel size of 5 µm × 5 µm in lipid analyses was increased to 25 µm × 25 µm for 4-ONE measurements to facilitate the detection of low-intensity signals. A single-ion-monitoring (SIM) experiment was performed first for 4-ONE, followed by a full MS scan.

SMALDI-MS images were created in Mirion[65] (TransMIT GmbH) using a bin width of $\Delta(m/z) = 0.004$; the images were normalized to total ion charge[66]. A digital mask created from a ubiquitous lipid signal was applied to the measurement area in order to exclude off-tissue pixels, and all images were stitched together in a single file to ensure uniform evaluation. An automatically generated list of all signals found in at least ten pixels in the stitched file was applied to the separate images to obtain the summed intensity of each signal. Signals were annotated in a bulk search against LIPID MAPS[67], allowing for [M + H]⁺, [M+Na]⁺, and [M + K]⁺ adducts and selecting the most likely lipid(s) for each measured mass. All annotations with a mass deviation <5 ppm were exported for further validation in HPLC MS² fragmentation experiments.

### HPLC MS² fragmentation

Approximately 1,300 rested and 1,300 sleep-deprived brains were collected in batches of 20–50 per session and snap frozen in plastic tubes. The frozen batches were combined in a glass Potter homogenizer, suspended in 50 µl ice-cold ammonium acetate (0.1% in water, Honeywell), manually homogenized, and transferred to a pre-cleaned Eppendorf tube. Lipids were extracted with 600 µl ice-cold methyl *tert*-butyl ether (MTBE, Sigma-Aldrich) and 150 µl methanol (VWR). After shaking the mixture for 1 h at 4 °C, 200 µl water (VWR) was added, the mixture was shaken for another 10 min, and the organic phase was collected after centrifugation for 5 min at 1,000g. The aqueous phase was re-extracted using an additional 400 µl MTBE, 120 µl methanol, and 100 µl water. The organic phases from both extraction steps were combined, and the solvent was evaporated under a stream of nitrogen for 30 min, leaving ~700 µg and ~800 µg of dry extract of rested and sleep-deprived samples, respectively. The extracts were stored at −80 °C until further use. An extraction blank was created by performing these steps without brain tissue.

Lipid extracts were thawed, dissolved in 650 µl acetonitrile, 300 µl isopropanol, and 50 µl water (all VWR) in an ultrasonic bath, and separated on a C18 column (100 mm × 2.1 mm, 2.6 µm particle size, 100 Å pore size; Phenomenex) in an UltiMate 3000 Rapid Separation System (ThermoFisher) coupled to an orbital trapping mass spectrometer (Q Exactive HF-X, ThermoFisher) using a heated electrospray ionization source (HESI II, ThermoFisher). Data-dependent acquisition and MS² fragmentation experiments were based on the inclusion list obtained from SMALDI-MSI annotations, with [M + H]⁺, [M+Na]⁺, [M + K]⁺ and [M + NH₄]⁺ adducts in positive-ion mode. Since the ionization mechanisms of MALDI and electrospray MS differ, MS² fragmentation of lipid extracts was additionally performed in negative-ion mode, considering [M−H]⁻ and [M + CHO₂]⁻ adducts, to increase the molecular coverage of SMALDI-MSI hits. Lipids were identified using LipidMatch[68]. All MS²-verified lipid annotations were validated by accurate mass and

the detection of all fatty acids plus the head group. Only one annotation (PE 27:2) was based on accurate mass and head group alone.

## Electrophysiology

Adult flies aged 2–6 days post eclosion were head-fixed to a custom mount using eicosane (Sigma). Cuticle, trachea, excess adipose tissue, and the perineural sheath were removed to create a small window, and the brain was continuously superfused with extracellular solution equilibrated with 95% $O_2$–5% $CO_2$ and containing (in mM) 103 NaCl, 3 KCl, 5 TES, 8 trehalose, 10 glucose, 7 sucrose, 26 $NaHCO_3$, 1 $NaH_2PO_4$, 1.5 $CaCl_2$, 4 $MgCl_2$, pH 7.3, 275 mOsM. GFP-positive cells were visualized on a Zeiss Axioskop 2 FS mot microscope equipped with a 60×/1.0 NA water-immersion objective (LUMPLFLN60XW, Olympus) and a pE-300 white LED light source (CoolLED). Borosilicate glass electrodes (9–11 MΩ for dFBNs, 5–7 MΩ for neurons of the pars intercerebralis) were fabricated on a PC-10 micropipette puller (Narishige) or a DMZ Universal Electrode Puller (Zeitz) and filled with intracellular solution containing (in mM) 10 HEPES, 140 potassium aspartate, 1 KCl, 4 MgATP, 0.5 $Na_3GTP$, 1 EGTA, pH 7.3, 265 mOsM. Where indicated, 50 µM 4-ONE or 200 µM 4-hydroxynonenal (4-HNE, Cayman Chemical) were added directly to the intracellular solution; in recordings from neurons of the pars intercerebralis, during which larger-diameter electrodes were used than in recordings from dFBNs, the 4-ONE concentration was lowered to 1 µM. Stock solutions of 4-ONE and 4-HNE were prepared in methyl acetate and ethanol, respectively; vehicle concentrations were not allowed to surpass 0.15% of the total volume after dilution. Recordings were obtained at room temperature with a MultiClamp 700B amplifier, lowpass-filtered at 10 kHz, and sampled at 20 or 50 kHz using Digidata 1440A or 1550B digitizers controlled through pCLAMP 10 or 11 (Molecular Devices). For photostimulation of miniSOG during whole-cell recordings[9], a 455-nm LED (Thorlabs M455L3) with a mounted collimator lens (Thorlabs ACP2520-A) and T-Cube LED Driver (Thorlabs) delivered 3.5–5 mW cm$^{-2}$ of optical power to the sample. Data were analysed using the NeuroMatic package[69] (http://neuromatic.thinkrandom.com) in Igor Pro (WaveMetrics).

Whole-cell capacitance compensation and bridge balance were used in voltage- and current-clamp recordings, respectively. Series resistances were monitored but not compensated and allowed to rise at most 20% above baseline—but never beyond 50 MΩ—during a recording. Uncompensated mean series resistances of ~40 MΩ in dFBNs (Extended Data Fig. 3c) caused predicted voltage errors of ~16 mV at typical $I_A$ amplitudes of ~400 pA (Extended Data Figs. 3a,b, 4 and 6). Input resistances were calculated from linear fits of the steady-state voltage changes elicited by 1-s steps of hyperpolarizing current (5-pA increments) from a pre-pulse potential of −60 ± 5 mV. Membrane time constants were estimated by fitting a single exponential to the voltage deflection caused by a hyperpolarizing 5-pA current step lasting 200 ms. Voltage-spike frequency functions were determined from voltage responses to a series of depolarizing current steps from a membrane potential of −60 ± 5 mV. To account for variations in input resistance within the dFBN population, the current required to produce a 5-mV hyperpolarizing voltage deflection from a pre-pulse potential of −60 ± 5 mV was used as a cell-specific unitary current step instead of a static 5-pA increment. Spikes were detected by finding minima in the time derivative of the membrane potential trace.

Voltage-clamp experiments on dFBNs and neurons of the pars intercerebralis were performed in the presence of 1 µM tetrodotoxin (Tocris) and 200 µM cadmium to block sodium and calcium currents, respectively. Potassium currents were measured by stepping neurons from holding potentials of −10 or −110 mV for 400 ms to a series of test potentials spanning the range from −100 mV to +30 mV in 10-mV increments[9,24]. Depolarizations from −110 mV produced the sum total of the cell's potassium currents ($I_{total}$, Extended Data Fig. 2a), whereas currents evoked by voltage steps from a holding potential of −10 mV lacked the $I_A$ (A-type or fast outward) component because voltage-gated

potassium channels such as Shaker inactivated (Extended Data Fig. 2b). $I_A$ was calculated by subtracting this non-A-type component from $I_{total}$ (Extended Data Fig. 2c). To determine the fast and slow inactivation time constants[9], double-exponential functions were fit to the decaying phase of A-type currents elicited by 400-ms steps to +30 mV (Extended Data Fig. 2d). In cases where the fits of slow inactivation time constants were poorly constrained, only the fast inactivation time constants were included in the analysis. Spiking was simulated by 3-ms depolarizing pulses to +10 mV, repeated at 10 Hz for 20 min.

Steady-state activation parameters were determined by applying depolarizing 400-ms voltage pulses from holding potentials of −10 or −110 mV; the pulses covered the range from −60 to +60 mV in steps of 10 mV. Linear leak currents were estimated from the slope of the current-voltage relationship at hyperpolarized potentials and subtracted. Steady-state inactivation parameters were obtained with the help of a two-pulse protocol, in which a 300-ms pre-pulse (−120 to +60 mV in 10-mV increments) was followed by a 400-ms test pulse to +30 mV; non-inactivating outward currents, measured from a pre-pulse potential of +10 mV, were subtracted. Peak A-type currents ($I_A$) were normalized to the maximum current amplitude ($I_{max}$) of the respective cell and plotted against the test or pre-pulse potentials ($V$). An estimated liquid junction potential[70] of 16.1 mV was subtracted post hoc. Curves were fit to the Boltzmann function $I_A/I_{max} = 1/\left(1 + e^{\frac{V - V_{0.5}}{k}}\right)$ to determine the half-maximal activation and inactivation voltages ($V_{0.5}$) and slope factors ($k$).

HEK-293 cells (CRL-1573, American Type Culture Collection) were grown at 37 °C under 5% $CO_2$ in Dulbecco's modified Eagle's medium (DMEM) with 10% (v/v) fetal bovine serum and 100 U ml$^{-1}$ penicillin plus 100 µg ml$^{-1}$ streptomycin (ThermoFisher). The cells were neither externally authenticated nor routinely tested for mycoplasma contamination. Cells were transfected (Lipofectamine 3000, ThermoFisher) with a 1:1 mixture of CMV promoter-driven expression vectors encoding mouse $K_V1.4$ and a bicistronic mouse $K_V\beta2$–IRES2–EGFP cassette. A carbonyl-reactive residue[71] (Cys-13) in the N-terminal inactivation peptide of $K_V1.4$ was mutated to serine. The growth medium was replaced during whole-cell recordings with extracellular solution containing (in mM) 10 HEPES, 140 NaCl, 5 KCl, 10 glucose, 2 $CaCl_2$, 1 $MgCl_2$, pH 7.4. Where indicated, HEK-293 cells were pre-incubated in extracellular solution supplemented with 12 mM methylglyoxal[19] for 1 h, followed by three washes with methylglyoxal-free solution, before data acquisition. GFP-positive cells were visually targeted with borosilicate glass electrodes (2–3 MΩ) filled with intracellular solution containing (in mM) 10 HEPES, 80 potassium aspartate, 60 KCl, 10 glucose, 2 MgATP, 1 $MgCl_2$, 5 EGTA, pH 7.3. Signals were acquired at room temperature with a MultiClamp 700B amplifier, lowpass-filtered at 10 kHz, and sampled at 20 kHz using a Digidata 1440 A digitizer controlled through pCLAMP 10 (Molecular Devices). Because untransfected HEK-293 cells lack voltage-gated conductances (Extended Data Fig. 7f), no channel blockers were present. To determine the fast and slow inactivation time constants, double-exponential functions were fit to the decaying phase of A-type currents elicited by 1-s steps to +30 mV. Spiking was simulated by 3-ms depolarizing pulses to +10 mV, repeated at 10 Hz for 20 min. Data were analysed using the NeuroMatic package[69] (http://neuromatic.thinkrandom.com) in Igor Pro (WaveMetrics).

## Confocal imaging

Dissected brains were fixed for 20 min in PBS with 4% (w/v) paraformaldehyde, washed 3 times for 20 min with 0.5% (v/v) Triton X-100 in PBS (PBST), and incubated sequentially at 4 °C in blocking solution (10% goat serum in PBST) overnight, with mouse monoclonal anti-Flag M2 antibodies (1:500, Sigma) in blocking solution for 2 days, and with goat anti-Mouse IgG Alexa Fluor 633 antibodies (1:500, ThermoFisher) for one day. The samples were washed 5 times with blocking solution before and after the addition of the secondary antibody, mounted in

Vectashield, and imaged on a Leica TCS SP5 confocal microscope with an HCX IRAPO L 25×/0.95 water-immersion objective.

## Statistics and reproducibility

With the exception of sleep measurements, no statistical methods were used to predetermine sample sizes. Flies of the indicated genotype, sex and age were selected randomly for analysis and assigned randomly to treatment groups if treatments were applied (for example, sleep deprivation). The investigators were not blinded to group allocation.

SMALDI-MSI signal intensities were analysed in LipidSig[72] and MATLAB (The MathWorks). Global differences between normalized glycerophospholipid intensities in cryosections of rested and sleep-deprived brains were evaluated by multiple $t$-tests with FDR-adjusted $P < 0.05$, using the method of Benjamini–Hochberg. Statistical associations with sleep history of user-defined lipid features, such as the indicated double-bond equivalent ranges or phospholipid head groups, were computed by Fisher's exact test in LipidSig[72]. Principal component and hierarchical cluster analyses were performed in MATLAB. The list of significantly different signals was exported and re-imported into Mirion to generate SMALDI-MS images for display. Behavioural and electrophysiological data were analysed in Prism 10 (GraphPad).

All null hypothesis tests were two-sided. To control type I errors, $P$ values were adjusted to achieve a joint $\alpha$ of 0.05 at each level in a hypothesis hierarchy; multiplicity adjusted $P$ values are reported in cases of multiple comparisons at one level. Group means or their time courses were compared by paired $t$-test, one- or two-way repeated-measures ANOVA, or mixed-effects models in cases where a variable was not measured in all cells at all time points, as indicated in figure legends. Repeated-measures ANOVA and mixed-effect models used the Geisser–Greenhouse correction in all instances except the comparisons of >2 genotypes in Fig. 3a,b and Extended Data Fig. 1a and were followed by planned pairwise analyses with Holm–Šídák's multiple comparisons test. Where the assumption of normality was violated (as indicated by D'Agostino–Pearson test), group means were compared by Mann–Whitney test, Wilcoxon test, Kruskal–Wallis ANOVA or Friedman test, followed by Dunn's multiple comparisons test to evaluate planned pairwise differences. Test statistics, degrees of freedom, and exact $P$ values are given in Supplementary Tables 1 and 2.

## Reporting summary

Further information on research design is available in the Nature Portfolio Reporting Summary linked to this article.

## Data availability

The SMALDI-MSI and LC-MS[2] datasets are accessible in METASPACE (https://metaspace2020.eu/project/drosophila and https://metaspace2020.eu/project/drosophila4ONE) and the MassIVE repository (ftp://massive.ucsd.edu/v05/MSV000091767/), respectively. All other data generated and analysed during this study are included in the Source Data.

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

**Acknowledgements** The authors thank L. Ballenberger and C. Hartmann for help with dissections and D. Anderson, B. Dickson, B. Ganetzky, T. Holmes, H. Jacobs, J. Ng, G. Rubin, S. Schneuwly, J. Simpson, the Bloomington Stock Center and the Vienna *Drosophila* Resource Center for flies. This work was supported by grants from the European Research Council (832467) and the UK Medical Research Council (MR/V013238/1) to G.M., and from the German Research Foundation (Sp314/23-1, INST 162/500-1 FUGG) and the Hessian Ministry of Science and Education (LOEWE Center DRUID) to B.S.; H.O.R. and L.G.S. received doctoral training fellowships from Wellcome and La Caixa, respectively; M.A.M. was supported by a Kekulé fellowship from the German Fonds der Chemischen Industrie; P.Z.L. was a Marshall Scholar; and A.K. held postdoctoral fellowships from the Swiss National Science Foundation and EMBO.

**Author contributions** H.O.R. performed all electrophysiological and behavioural experiments on flies and M.A.M. performed all lipidomic analyses, under the supervision of S.G. and B.S., on material prepared by L.G.S., H.O.R. and A.K. P.Z.L. characterized *Hk*[Flag] flies and K$_v$ currents in HEK-293 cells. B.S. designed the SMALDI-MSI methodology and instrumentation. G.M. devised and directed the research and wrote the paper.

**Competing interests** M.A.M. and S.G. are employees of and B.S. is a consultant for TransMIT GmbH. The other authors declare no competing interests.

**Additional information**
**Correspondence and requests for materials** should be addressed to Gero Miesenböck.

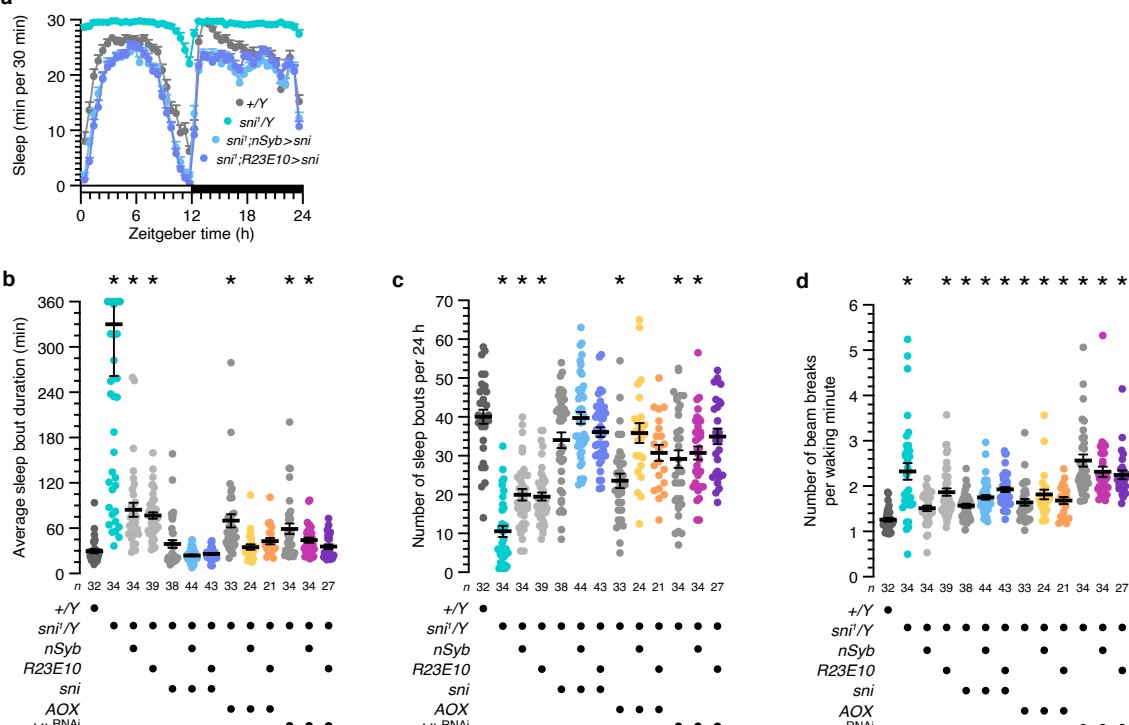

**Extended Data Fig. 1 | Sleep architecture and waking locomotor activity of *sni¹* mutants. a**, The *nSyb-GAL4-* or *R23E10-GAL4*-driven overexpression of sniffer (over)corrects the altered sleep profile of hemizygous *sni¹* mutant males ($P < 0.0001$ for all pairwise comparisons, two-way repeated-measures ANOVA with Holm-Šídák test; sample sizes in **b**). **b**, The average sleep bout duration in hemizygous *sni¹* mutant males differs from wild-type ($P < 0.0001$; Kruskal-Wallis ANOVA with Dunn's test) but returns to control level if carriers also express sniffer or AOX pan-neuronally under the control of *nSyb-GAL4* (sni: $P > 0.9999$; AOX: $P > 0.9999$) or sniffer, AOX, or $Hk^{RNAi}$ in dFBNs under the control of *R23E10-GAL4* (sni: $P > 0.9999$; AOX: $P = 0.1462$; $Hk^{RNAi}$: $P > 0.9999$). The average sleep bout durations of 6 *sni¹* mutants exceeding 360 min are plotted at the top

of the graph; mean and s.e.m. are based on the actual values. **c**, The number of sleep bouts in hemizygous *sni¹* mutant males differs from wild-type ($P < 0.0001$; Kruskal-Wallis ANOVA with Dunn's test) but returns to control level if carriers also express sniffer or AOX pan-neuronally under the control of *nSyb-GAL4* (sni: $P > 0.9999$; AOX: $P > 0.9999$) or sniffer, AOX, or $Hk^{RNAi}$ in dFBNs under the control of *R23E10-GAL4* (sni: $P > 0.9999$; AOX: $P = 0.1492$; $Hk^{RNAi}$: $P > 0.9999$). **d**, Hemizygous *sni¹* mutant males show elevated waking locomotor activity relative to wild-type ($P < 0.0001$; Kruskal-Wallis ANOVA with Dunn's test). Data are means ± s.e.m.; *n*, number of flies; asterisks, significant differences ($P < 0.05$) from wild-type in planned pairwise comparisons. For statistical details see Supplementary Table 2.

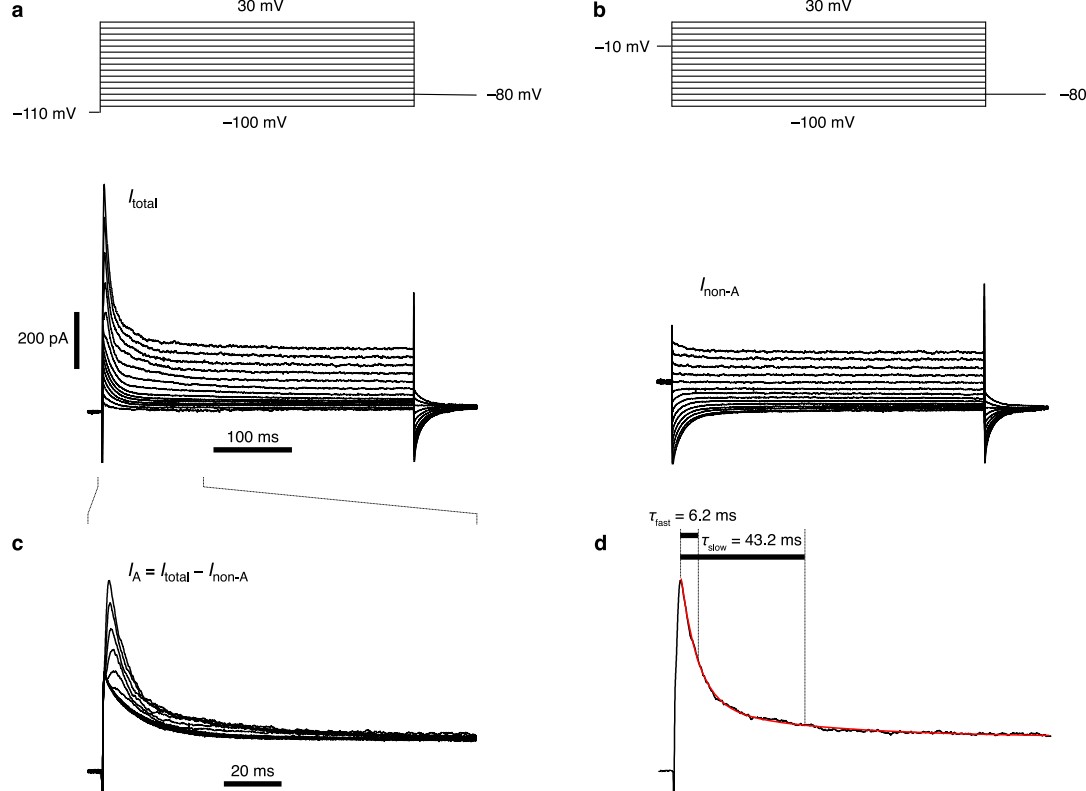

**Extended Data Fig. 2 | Measurement of the inactivation time constants of $I_A$. a**, Voltage steps from a holding potential of –110 mV (top) elicit the full complement of potassium currents in a dFBN ($I_{total}$, bottom). **b**, Stepping the same neuron from a holding potential of –10 mV (top) elicits potassium currents lacking the A-type component ($I_{non-A}$, bottom). **c**, Digital subtraction of $I_{non-A}$ (**b**, bottom) from $I_{total}$ (**a**, bottom) yields $I_A$. Note the expanded timescale. **d**, Estimates of $\tau_{fast}$ and $\tau_{slow}$ are obtained from a double-exponential fit (red line) to the A-type current evoked by step depolarization to +30 mV.

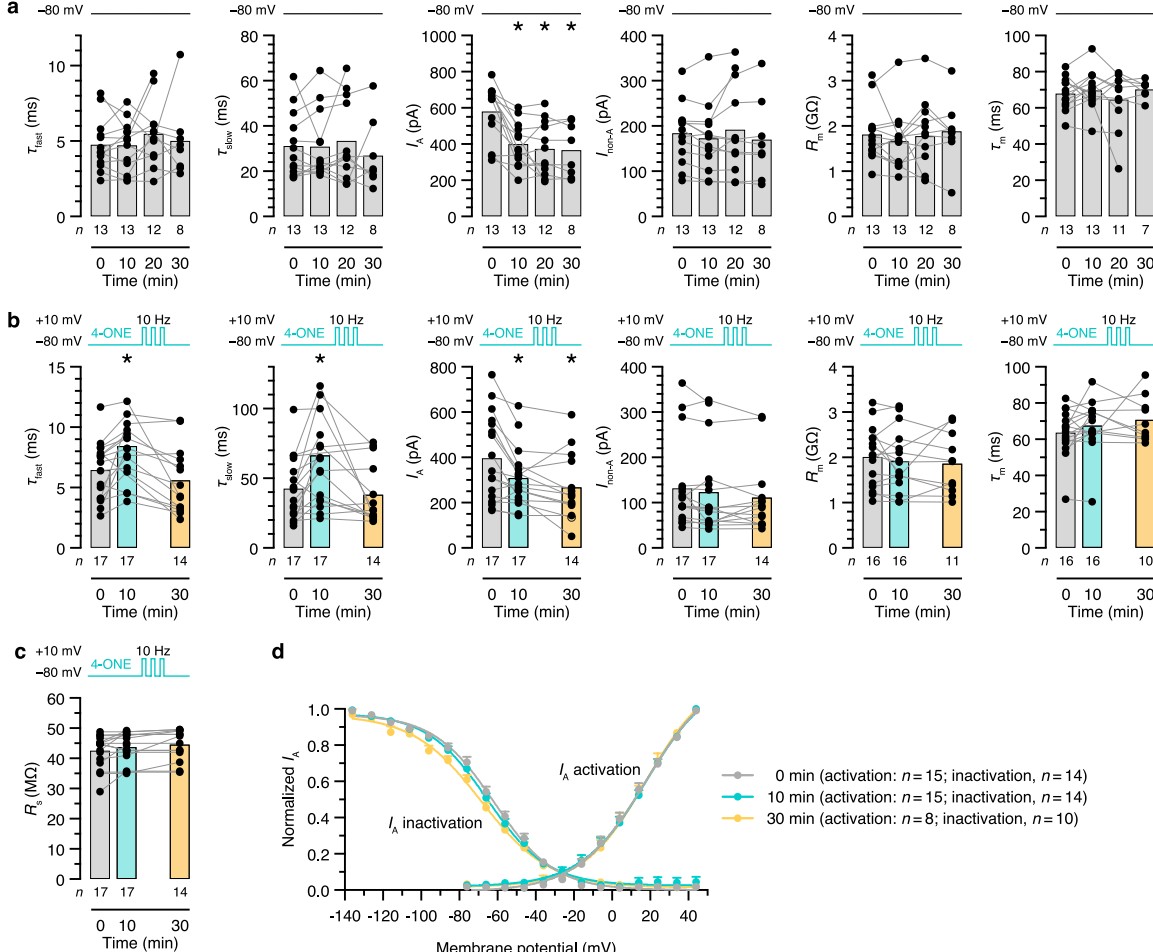

**Extended Data Fig. 3 | Inactivation kinetics and amplitudes of potassium currents, series resistances, and steady-state activation and inactivation curves of $I_A$ during the course of a 30-minute recording. a**, dFBNs were held at −80 mV, except during the voltage protocols required to measure $I_A$. In the absence of 4-ONE, the fast and slow inactivation time constants of $I_A$ ($\tau_{fast}$: $P = 0.2499$; $\tau_{slow}$: $P = 0.5968$; mixed-effects model), the amplitude of $I_{non-A}$ ($P = 0.3527$; mixed-effects model), input resistance ($P = 0.6543$; mixed-effects model), and membrane time constant ($P = 0.5196$; mixed-effects model) remain unchanged, but the amplitude of $I_A$ runs down during the course of the recording ($P = 0.0004$; mixed-effects model). Columns, population averages; dots, individual cells; $n$, number of cells; asterisks, significant differences ($P < 0.05$) relative to baseline in planned pairwise comparisons. **b**, **c**, dFBNs were held at −80 mV in the interval of 0–10 min (except during the voltage protocols required to measure $I_A$) and repeatedly step-depolarized to +10 mV (3 ms, 10 Hz) between 10 and 30 min. The inclusion of 50 µM 4-ONE in the intracellular solution increases the fast and slow inactivation time constants of $I_A$ above the baselines recorded immediately after break-in (**b**, turquoise vs. grey shading); a series of depolarization steps between 10 and 30 min

counteracts this increase despite the continuous presence of 4-ONE (**b**, yellow shading; $\tau_{fast}$: $P = 0.0054$; $\tau_{slow}$: $P = 0.0014$; mixed-effects model). The amplitude of $I_A$ runs down during the course of the recording (**b**, $P = 0.0008$; mixed-effects model); $I_{non-A}$ (**b**, $P = 0.3120$; mixed-effects model), input resistance (**b**, $P = 0.4961$; mixed-effects model), and membrane time constant (**b**, $P = 0.2282$; mixed-effects model) remain unchanged. Series resistance increases gradually (**c**, $P = 0.0399$; mixed-effects model) but remains within <20% of baseline and below 50 MΩ. Columns, population averages; dots, individual cells; $n$, number of cells; asterisks, significant differences ($P < 0.05$) relative to the 0-minute time point in planned pairwise comparisons by Holm-Šídák test. **d**, Steady-state activation and inactivation curves of $I_A$ in dFBNs immediately after break-in (0 min), after 10 min of dialysis with intracellular solution containing 50 µM 4-ONE (turquoise), and after a series of depolarization steps to +10 mV (3 ms, 10 Hz) between 10 and 30 min (yellow). Data are means ± s.e.m; solid lines, Boltzmann fits. The half-activation voltages and activation slope factors are identical at all time points ($P = 0.5378$, $F$ test) but the half-inactivation voltages and inactivation slope factors differ ($P < 0.0001$, $F$ test). For statistical details see Supplementary Table 2.

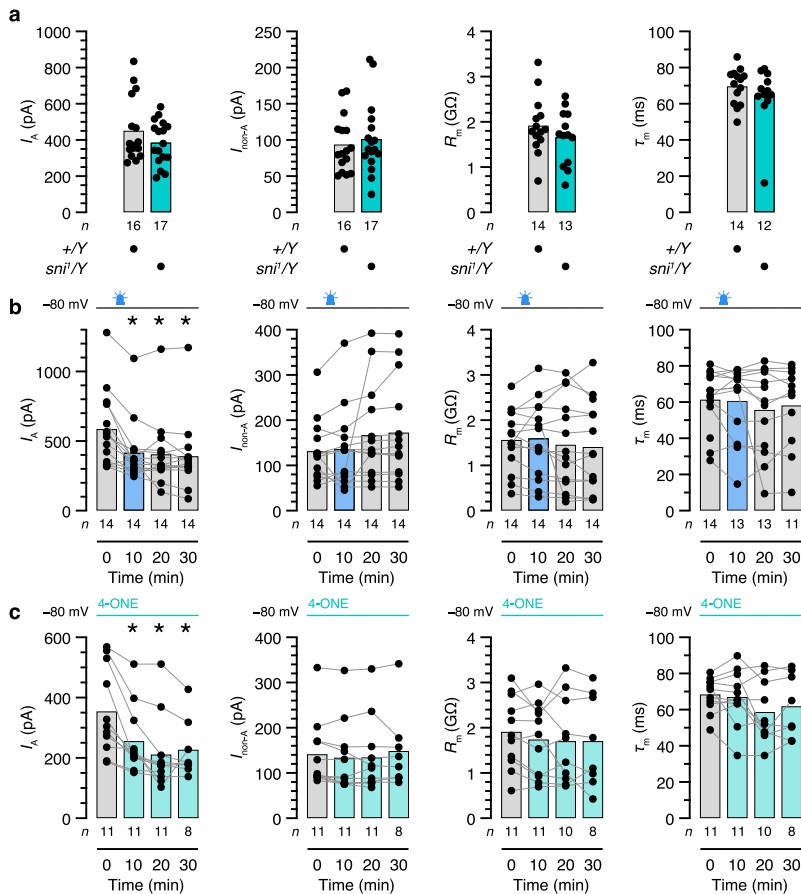

**Extended Data Fig. 4 | Potassium current amplitudes and membrane properties of dFBNs in Fig. 4. a**, dFBNs of hemizygous $sni^I$ mutant (turquoise) and wild-type males (grey) do not differ with respect to the amplitudes of $I_A$ ($P = 0.4023$, two-sided Mann-Whitney test) and $I_{non-A}$ ($P = 0.6276$, two-sided $t$-test), input resistance ($P = 0.3014$, two-sided $t$-test), and membrane time constant ($P = 0.5267$, two-sided Mann-Whitney test). **b**, dFBNs expressing miniSOG were held at −80 mV, except during the voltage protocols required to measure $I_A$, and exposed to blue light between the 0- and 10-minute time points (blue shading). The amplitude of $I_A$ runs down during the course of the recording ($P = 0.0003$; Friedman test); $I_{non-A}$ ($P = 0.1116$; Friedman test), input resistance ($P = 0.4361$; repeated-measures ANOVA), and membrane time

constant ($P = 0.3265$; mixed-effects model) remain unchanged. **c**, dFBNs were held at −80 mV, except during the voltage protocols required to measure $I_A$, and dialyzed with 50 μM 4-ONE (turquoise shading). The amplitude of $I_A$ runs down during the course of the recording ($P = 0.0024$; mixed-effects model); $I_{non-A}$ ($P = 0.2067$; mixed-effects model), input resistance ($P = 0.2942$; mixed-effects model), and membrane time constant ($P = 0.0783$; mixed-effects model) remain unchanged. Columns, population averages; dots, individual cells; $n$, number of cells; asterisks, significant differences ($P < 0.05$) relative to the 0-minute time point in planned pairwise comparisons by Holm-Šídák or Dunn's test. For statistical details see Supplementary Table 2.

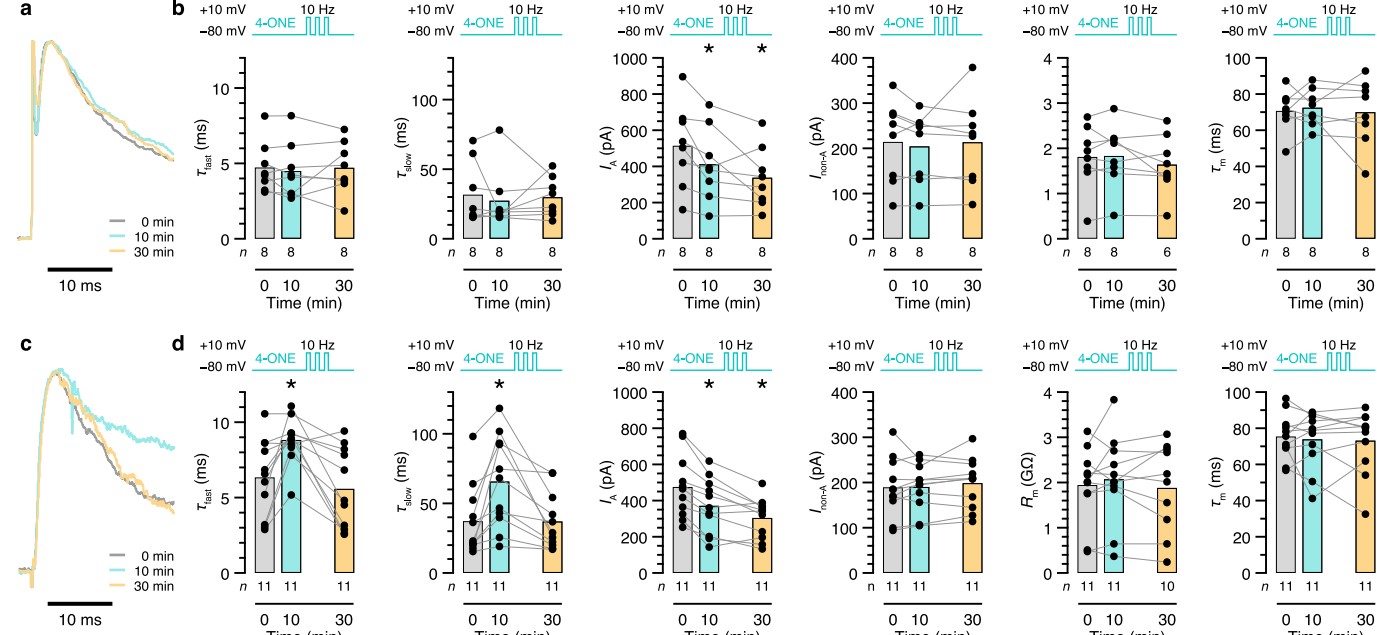

**Extended Data Fig. 5 | Memory storage and erasure requires catalytically active $K_V\beta$. a, b**, dFBNs expressing a catalytically defective *Hk* (K289M) 'rescue' transgene in a homozygous *Hk^1* mutant background. The cells were held at −80 mV between 0 and 10 min (except during the voltage protocols required to measure $I_A$) and repeatedly step-depolarized to +10 mV (3 ms, 10 Hz) between 10 and 30 min. The inclusion of 50 µM 4-ONE in the intracellular solution fails to increase the fast and slow inactivation time constants of $I_A$ above the baselines recorded immediately after break-in (**b**, turquoise vs. grey shading); a series of depolarization steps between 10 and 30 min is similarly without effect (**b**, yellow shading; $\tau_{fast}$: $P = 0.6841$, repeated-measures ANOVA; $\tau_{slow}$: $P = 0.7852$, Friedman test; examples of peak-normalized $I_A$ evoked in the same dFBN by voltage steps to +30 mV in **a**). The amplitude of $I_A$ runs down during the course of the recording (**b**, $P = 0.0087$; repeated-measures ANOVA); $I_{non-A}$ (**b**, $P = 0.6730$; repeated-measures ANOVA), input resistance (**b**, $P = 0.2615$; repeated-measures ANOVA), and membrane time constant (**b**, $P = 0.8143$; repeated-measures ANOVA) remain unchanged. **c, d**, dFBNs expressing a catalytically competent *Hk* rescue transgene in a homozygous *Hk^1* mutant background. The cells were

held at −80 mV between 0 and 10 min (except during the voltage protocols required to measure $I_A$) and repeatedly step-depolarized to +10 mV (3 ms, 10 Hz) between 10 and 30 min. The inclusion of 50 µM 4-ONE in the intracellular solution increases the fast and slow inactivation time constants of $I_A$ above the baselines recorded immediately after break-in (**d**, turquoise vs. grey shading); a series of depolarization steps between 10 and 30 min counteracts this increase despite the continuous presence of 4-ONE (**d**, yellow shading; $\tau_{fast}$: $P = 0.0020$; $\tau_{slow}$: $P < 0.0001$; Friedman test; examples of peak-normalized $I_A$ evoked in the same dFBN by voltage steps to +30 mV in **c**). The amplitude of $I_A$ runs down during the course of the recording (**d**, $P < 0.0001$; repeated-measures ANOVA); $I_{non-A}$ (**d**, $P = 0.4334$; repeated-measures ANOVA), input resistance (**d**, $P = 0.5984$; mixed-effects model), and membrane time constant (**d**, $P = 0.9761$; Friedman test) remain unchanged. Columns, population averages; dots, individual cells; $n$, number of cells; asterisks, significant differences ($P < 0.05$) relative to the 0-minute time point in planned pairwise comparisons by Holm-Šídák or Dunn's test. For statistical details see Supplementary Table 2.

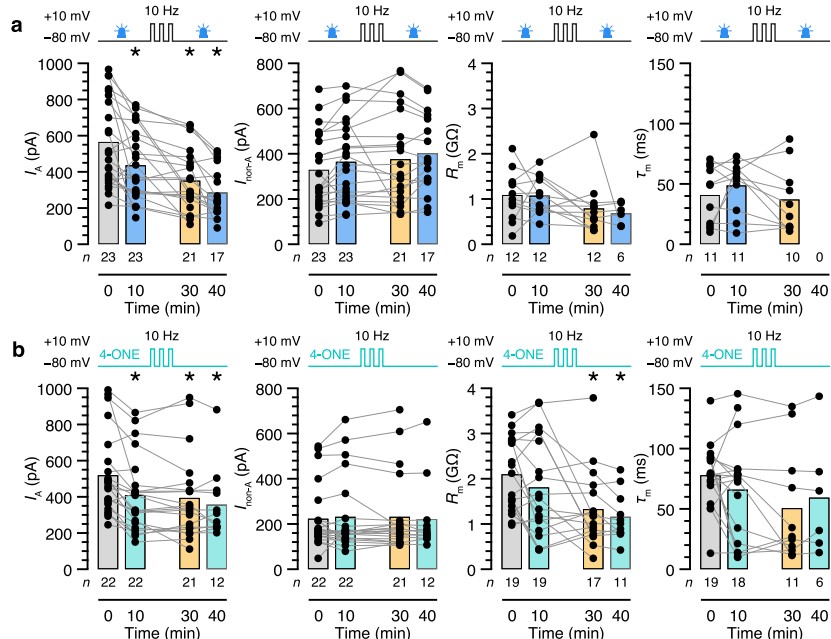

**Extended Data Fig. 6 | Potassium current amplitudes and membrane properties of dFBNs in Fig. 6. a**, dFBNs expressing miniSOG were held at −80 mV in the intervals of 0–10 and 30–40 min (except during the voltage protocols required to measure $I_A$) and repeatedly step-depolarized to +10 mV (3 ms, 10 Hz) between 10 and 30 min. Nine-minute exposures to blue light (between the 0- and 10-minute and the 30- and 40-minute time points) leave $I_{non-A}$ ($P = 0.1673$; mixed-effects model), input resistance ($P = 0.0688$; mixed-effects model), and membrane time constant ($P = 3058$; mixed-effects model) unchanged, but the amplitude of $I_A$ runs down during the course of the recording ($P < 0.0001$, mixed-effects model). **b**, dFBNs were held at −80 mV in the intervals of 0–10 and 30–40 min (except during the voltage protocols

required to measure $I_A$) and repeatedly step-depolarized to +10 mV (3 ms, 10 Hz) between 10 and 30 min. The cells were dialyzed with 50 μM 4-ONE in the intracellular solution (turquoise shading). The amplitude of $I_A$ runs down during the course of the recording ($P < 0.0001$; mixed-effects model); input resistance decreases after the series of depolarization steps ($P = 0.0008$; mixed-effects model); $I_{non-A}$ ($P = 0.6240$; mixed-effects model) and membrane time constant ($P = 0.1258$; mixed-effects model) remain unchanged. Columns, population averages; dots, individual cells; $n$, number of cells; asterisks, significant differences ($P < 0.05$) relative to the 0-minute time point in planned pairwise comparisons by Holm-Šídák test. For statistical details see Supplementary Table 2.

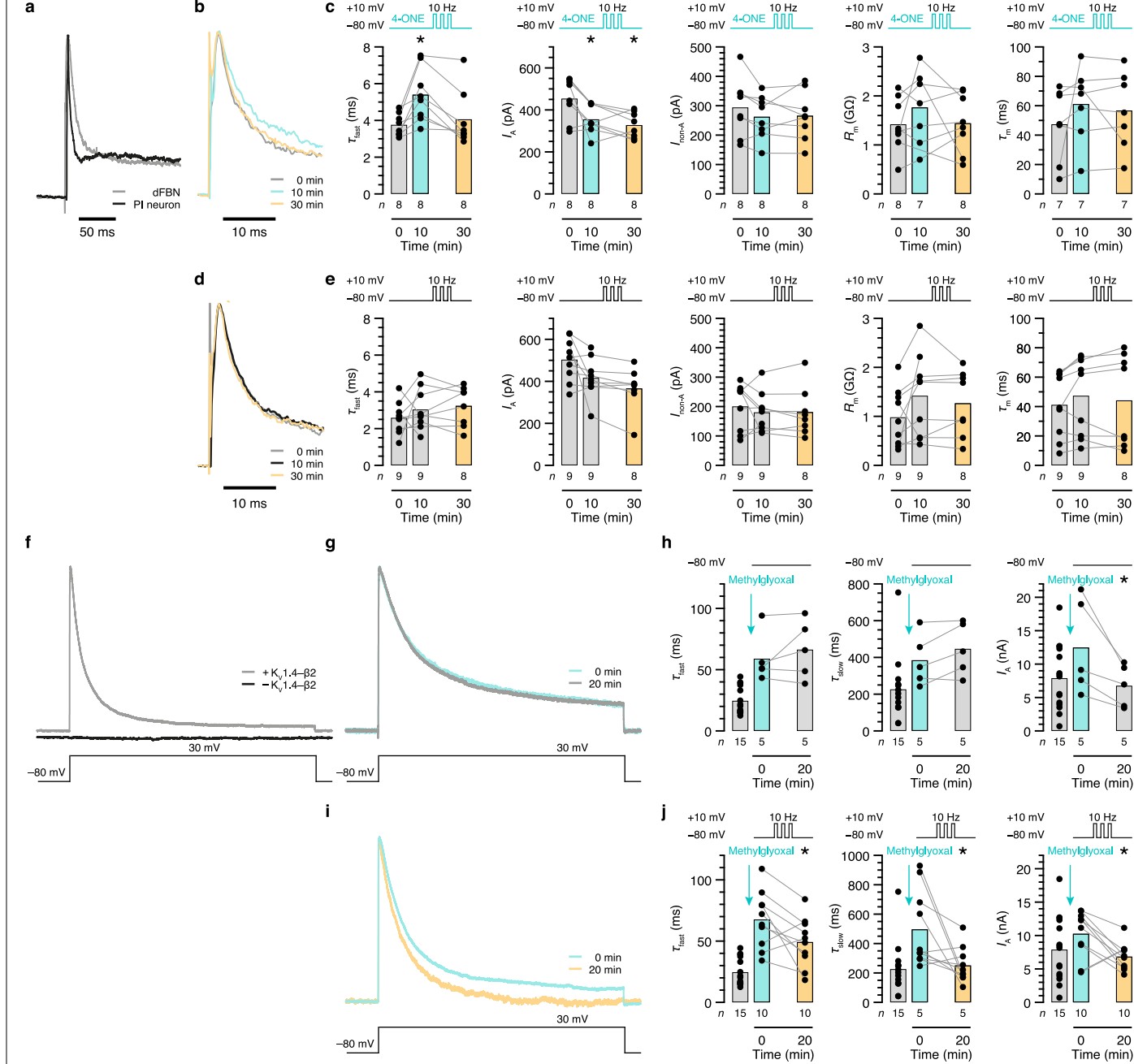

**Extended Data Fig. 7** | See next page for caption.

**Extended Data Fig. 7 | Lipid peroxidation products alter the inactivation kinetics of $I_A$ in non-dFB neurons and cultured cells expressing mammalian $K_V1.4$ and $K_V\beta2$. a**, Examples of peak-normalized transmembrane currents evoked by 1-s voltage pulses from −80 mV to +30 mV in a dFBN (grey) and a neuron of the pars intercerebralis (PI neuron) (black). A slowly activating outward current in the PI neuron interferes with an accurate measurement of $\tau_{slow}$. **b**, **c**, PI neurons were held at −80 mV between 0 and 10 min (except during the voltage protocols required to measure $I_A$) and repeatedly step-depolarized to +10 mV (3 ms, 10 Hz) between 10 and 30 min. The inclusion of 1 µM 4-ONE in the intracellular solution increases the fast inactivation time constant of $I_A$ above the baseline measured immediately after break-in (**c**, turquoise vs. grey shading); a series of depolarization steps between 10 and 30 min counteracts this increase despite the continuous presence of 4-ONE (**c**, yellow shading; $P = 0.0009$; Friedman test; examples of peak-normalized $I_A$ evoked in the same PI neuron by voltage steps to +30 mV in **b**). The amplitude of $I_A$ runs down during the course of the recording (**c**, $P = 0.0075$; repeated-measures ANOVA); $I_{non-A}$ (**c**, $P = 0.1035$; repeated-measures ANOVA), input resistance (**c**, $P = 0.4532$; mixed-effects model), and membrane time constant (**c**, $P = 0.4861$; Friedman test) remain unchanged. **d**, **e**, PI neurons were held at −80 mV between 0 and 10 min (except during the voltage protocols required to measure $I_A$) and repeatedly step-depolarized to +10 mV (3 ms, 10 Hz) between 10 and 30 min. In the absence of 4-ONE, the fast inactivation time constant of $I_A$ (**e**, $P = 0.3416$; mixed-effects model; examples of peak-normalized $I_A$ evoked in the same PI neuron by voltage steps to +30 mV in **d**), the amplitude of $I_{non-A}$ (**e**, $P = 0.3712$; mixed-effects model), input resistance (**e**, $P = 0.1304$; mixed-effects model), and membrane time constant (**e**, $P = 0.2109$; mixed-effects model) remain unchanged, but the amplitude of $I_A$ runs down during the course of the recording ($P = 0.0496$; mixed-effects model). **f**, Examples of peak-normalized transmembrane currents evoked by 1-s voltage pulses from −80 mV to +30 mV in HEK-293 cells expressing mouse $K_V1.4$ and $K_V\beta2$ (grey), or in untransfected HEK-293 cells (black). **g–j**, HEK-293 cells expressing mouse $K_V1.4$ and $K_V\beta2$. A 1-h exposure to 12 mM methylglyoxal, followed by three washes with methylglyoxal-free solution, increases the fast and slow inactivation time constants of transmembrane currents relative to those of cells maintained in the absence of methylglyoxal (**h**, **j**, turquoise vs. grey shading; $\tau_{fast}$: $P < 0.0001$; $\tau_{slow}$: $P < 0.0001$; two-sided Mann-Whitney test). In cells held at −80 mV (except during the voltage protocols required to measure $I_A$), the time constants remain stably elevated for 20 min (**h**, $\tau_{fast}$: $P = 0.4375$; $\tau_{slow}$: $P = 0.1875$; two-sided Wilcoxon test; examples of peak-normalized currents in **g**), but a series of depolarization steps (3 ms, 10 Hz, 20 min) to +10 mV reverses the increase (**j**, yellow shading; $\tau_{fast}$: $P = 0.0294$, two-sided paired $t$-test; $\tau_{slow}$: $P = 0.0137$, two-sided Wilcoxon test; examples of peak-normalized currents in **i**). The amplitude of $I_A$ runs down during the course of the recording ($P = 0.0429$, two-sided paired $t$-test). Columns, population averages; dots, individual cells; $n$, number of cells; asterisks, significant differences ($P < 0.05$) relative to the 0-minute time point by Holm-Šídák or Dunn's test. For statistical details see Supplementary Table 2.

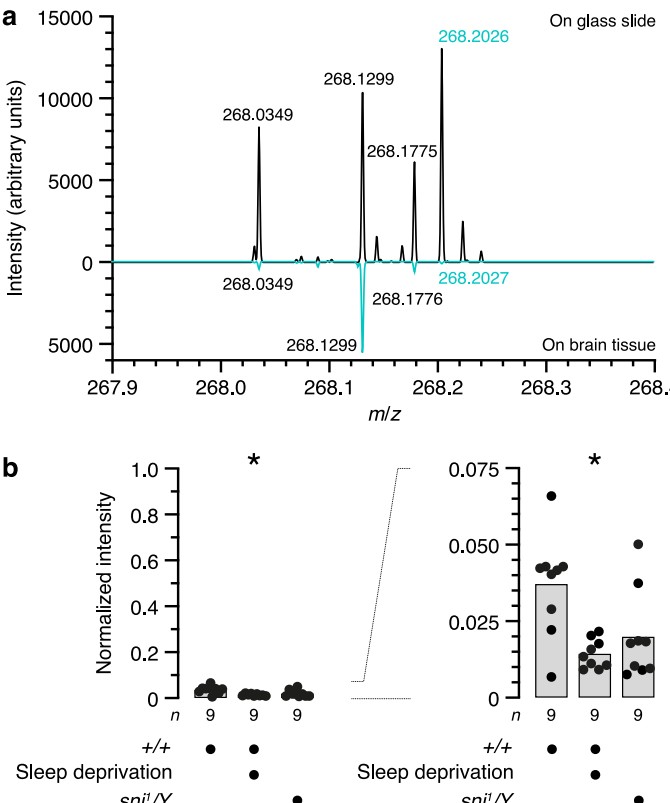

**Extended Data Fig. 8 | SMALDI-MSI analysis of 4-ONE. a**, Mirror plot of mass spectra of 100 µM 4-ONE standard on a blank slide (top) or a brain cryosection (bottom). Spectra were acquired in single-ion-monitoring mode at the calculated $m/z$ of the [4-ONE+GirT-H$_2$O]$^+$ ion (268.2020); peaks with a mass deviation <5 ppm are labelled in green type. **b**, The intensity of the [4-ONE+GirT-H$_2$O]$^+$ signal is decreased in cryosections of sleep-deprived brains ($P$ = 0.0179, Kruskal-Wallis ANOVA) but not significantly altered in hemizygous $sni^1$ mutant males ($P$ = 0.0560). Intensities on the left are normalized to a 100 µM 4-ONE standard on a blank slide; the scale is expanded on the right. Columns, population averages; dots, individual cryosections; $n$, number of cryosections; asterisks, significant differences ($P$ < 0.05) relative to rested wild-type flies in planned pairwise comparisons by Dunn's test. For statistical details see Supplementary Table 2.

# Reporting Summary

## Statistics

For all statistical analyses, confirm that the following items are present in the figure legend, table legend, main text, or Methods section.

| n/a | Confirmed | |
|---|---|---|
| ☐ | ☒ | The exact sample size (*n*) for each experimental group/condition, given as a discrete number and unit of measurement |
| ☐ | ☒ | A statement on whether measurements were taken from distinct samples or whether the same sample was measured repeatedly |
| ☐ | ☒ | The statistical test(s) used AND whether they are one- or two-sided *Only common tests should be described solely by name; describe more complex techniques in the Methods section.* |
| ☒ | ☐ | A description of all covariates tested |
| ☐ | ☒ | A description of any assumptions or corrections, such as tests of normality and adjustment for multiple comparisons |
| ☐ | ☒ | A full description of the statistical parameters including central tendency (e.g. means) or other basic estimates (e.g. regression coefficient) AND variation (e.g. standard deviation) or associated estimates of uncertainty (e.g. confidence intervals) |
| ☐ | ☒ | For null hypothesis testing, the test statistic (e.g. *F*, *t*, *r*) with confidence intervals, effect sizes, degrees of freedom and *P* value noted *Give P values as exact values whenever suitable.* |
| ☒ | ☐ | For Bayesian analysis, information on the choice of priors and Markov chain Monte Carlo settings |
| ☒ | ☐ | For hierarchical and complex designs, identification of the appropriate level for tests and full reporting of outcomes |
| ☒ | ☐ | Estimates of effect sizes (e.g. Cohen's *d*, Pearson's *r*), indicating how they were calculated |

*Our web collection on statistics for biologists contains articles on many of the points above.*

## Software and code

Policy information about availability of computer code

| Data collection | SMALDI-MSI: SMALDIControl 1.3 (TransMIT GmbH) and Tune 2.8 (ThermoFisher) <br> LC-MS2: Tune 2.9 (ThermoFisher) and Thermo XCalibur 4.0.27.19 (ThermoFisher) <br> Sleep behaviour: Trikinetics DAM system <br> Electrophysiology: pCLAMP 10 or 11 (Molecular Devices) <br> Confocal images: Leica Application Suite; summed-intensity projections were computed in Fiji 2.14.0/1.54f |
|---|---|
| Data analysis | SMALDI-MSI data were analysed in Mirion 3.3.64.23 (TransMIT GmbH) and annotated in bulk structure searches against the COMP_DB database in LIPID MAPS (https://www.lipidmaps.org). Signal intensities and degrees of enrichment of molecular features were analysed in MATLAB R2023b and the Differential Expression Analysis module of LipidSig (https://lipidsig.bioinfomics.org/DE/). <br> LC-MS2 data were annotated using LipidMatch 3.5 (Innovative Omics). <br> Sleep behaviour data were analysed with the Sleep and Circadian Analysis MATLAB Program (SCAMP v3). <br> Electrophysiological data were analysed using version 3.0c of the NeuroMatic package in Igor Pro 8.04 (WaveMetrics). <br> All normality and hypothesis tests were performed in Prism 10.4.1 (GraphPad). |

For manuscripts utilizing custom algorithms or software that are central to the research but not yet described in published literature, software must be made available to editors and reviewers. We strongly encourage code deposition in a community repository (e.g. GitHub). See the Nature Portfolio guidelines for submitting code & software for further information.

## Data

Policy information about availability of data

All manuscripts must include a data availability statement. This statement should provide the following information, where applicable:

- Accession codes, unique identifiers, or web links for publicly available datasets
- A description of any restrictions on data availability
- For clinical datasets or third party data, please ensure that the statement adheres to our policy

The SMALDI-MSI and LC-MS2 datasets are accessible in METASPACE (https://metaspace2020.eu/project/drosophila and https://metaspace2020.eu/project/drosophila4ONE) and the MassIVE repository (ftp://massive.ucsd.edu/v05/MSV000091767/), respectively. All other data generated and analysed during this study are included in the Source Data file.

## Research involving human participants, their data, or biological material

Policy information about studies with human participants or human data. See also policy information about sex, gender (identity/presentation), and sexual orientation and race, ethnicity and racism.

| Reporting on sex and gender | n/a |
|---|---|
| Reporting on race, ethnicity, or other socially relevant groupings | n/a |
| Population characteristics | n/a |
| Recruitment | n/a |
| Ethics oversight | n/a |

Note that full information on the approval of the study protocol must also be provided in the manuscript.

# Field-specific reporting

Please select the one below that is the best fit for your research. If you are not sure, read the appropriate sections before making your selection.

☒ Life sciences          ☐ Behavioural & social sciences          ☐ Ecological, evolutionary & environmental sciences

For a reference copy of the document with all sections, see nature.com/documents/nr-reporting-summary-flat.pdf

# Life sciences study design

All studies must disclose on these points even when the disclosure is negative.

| Sample size | Sample sizes are provided in each figure and extended data figure or its legend.<br>Sample sizes in behavioural experiments were chosen to detect 2-h differences in daily sleep with a power of 0.8.<br>Sample sizes in electrophysiological experiments are based on precedent (Kempf et al., Nature 2019).<br>Sample sizes in lipidomic experiments match those of example datasets in the Differential Expression Analysis module of LipidSig (https://lipidsig.bioinfomics.org/DE/). |
|---|---|
| Data exclusions | Immobile flies (< 2 beam breaks per 24 h) were excluded from sleep measurements.<br>Voltage-clamp recordings were terminated if the series resistance increased by >20% from baseline or exceeded 50 MΩ.<br>If fits of the slow inactivation time constants of A-type currents were poorly constrained, only the fast inactivation time constants were analysed. |
| Replication | Results were replicated on different brain sections, flies, or cells across each dataset. All replicates are included in figures and extended data figures. |
| Randomization | Flies of the correct genotype and sex, as indicated in Methods, were selected randomly for analysis and assigned randomly to treatment groups if treatments were applied (e.g., sleep deprivation). |
| Blinding | The investigators were not blind to group allocation. Measurements and analyses were automated and/or required the performance of genotype-specific experimental protocols. |

# Reporting for specific materials, systems and methods

We require information from authors about some types of materials, experimental systems and methods used in many studies. Here, indicate whether each material, system or method listed is relevant to your study. If you are not sure if a list item applies to your research, read the appropriate section before selecting a response.

## Materials & experimental systems

| n/a | Involved in the study |
|-----|----------------------|
| ☐ | ☒ Antibodies |
| ☐ | ☒ Eukaryotic cell lines |
| ☒ | ☐ Palaeontology and archaeology |
| ☐ | ☒ Animals and other organisms |
| ☒ | ☐ Clinical data |
| ☒ | ☐ Dual use research of concern |
| ☒ | ☐ Plants |

## Methods

| n/a | Involved in the study |
|-----|----------------------|
| ☒ | ☐ ChIP-seq |
| ☒ | ☐ Flow cytometry |
| ☒ | ☐ MRI-based neuroimaging |

## Antibodies

| | |
|---|---|
| Antibodies used | Mouse monoclonal anti-FLAG M2 antibody (Sigma F1804)<br>Goat anti-Mouse Alexa Fluor 633 antibody (ThermoFisher A-21052) |
| Validation | The mouse monoclonal anti-FLAG M2 antibody recognizes the artificial epitope DYKDDDDK, has been used in immunofluorescence applications in Drosophila (e.g., Tannan et al., PLoS Genet 2018), and shows no cross-reactivity with endogenous Drosophila proteins in our hands (Fig. 5f). |

## Eukaryotic cell lines

Policy information about cell lines and Sex and Gender in Research

| | |
|---|---|
| Cell line source(s) | HEK-293 cells (CRL-1573, American Type Culture Collection) |
| Authentication | Our sample of HEK-293 cells was not externally authenticated. |
| Mycoplasma contamination | Our sample of HEK-293 cells was not tested routinely for mycoplasma infection. |
| Commonly misidentified lines<br>(See ICLAC register) | No commonly misidentified cell lines were used in this study. |

## Animals and other research organisms

Policy information about studies involving animals; ARRIVE guidelines recommended for reporting animal research, and Sex and Gender in Research

| | |
|---|---|
| Laboratory animals | In all experiments except those involving the sni1 mutation and all relevant controls (see below), females aged 2–6 days after eclosion were used. These experimental flies were heterozygous for all transgenes and homozygous for either a wild-type or mutant (Hk1) Hyperkinetic allele. Transgenes included R23E10-GAL4 or Dh31-GAL4 and UAS–mCD8::GFP, UAS–myr-MS6T2, UAS–Hk, and/or UAS–HkK289M.<br>The function of the X-linked sniffer gene was studied in males aged 2–6 days after eclosion. Hemizygous carriers of the sni1 allele coexpressed UAS–mCD8::GFP, UAS–sni, UAS–AOX, or UAS–HkRNAi (47805GD) transgenes under the control of nSyb–GAL4 or R23E10-GAL4.<br>In the HkFLAG strain, the endogenous Hyperkinetic locus encodes an in-frame fusion with an N-terminal FLAG epitope. |
| Wild animals | No wild animals were used in this study. |
| Reporting on sex | Male flies were used in behavioural and electrophysiological analyses of sniffer mutants to facilitate the synthesis of the desired genotypes (the sniffer gene is X-linked). Female flies were used in all other experiments because of their larger body size. |
| Field-collected samples | No field-collected samples were used in this study. |
| Ethics oversight | Ethical approval was not required for this study. |

Note that full information on the approval of the study protocol must also be provided in the manuscript.

## Plants

Seed stocks

n/a

Novel plant genotypes

n/a

Authentication

n/a

