## [Peer Review File · Nature]

Sleep pressure accumulates in a voltage-gated lipid peroxidation memory

Corresponding Author: Professor Gero Miesenboeck

This file contains all reviewer reports in order by version, followed by all author rebuttals in order by version. Parts of this Peer Review File have been redacted as indicated to maintain the confidentiality of data.

Version 0:

Reviewer comments:

Referee #1

(Remarks to the Author)

In this study, the authors build on previous investigations proposing that the beta subunits of the Kv1 channels in drosophila(hyperkinetic) are important for sleep regulation via the status of its NADP+ cofactor which varies in response to mitochondrial ROS. Here they extend this model to propose that lipid peroxidation products, primarily 4-ONE mediate this communication and cause formation of NADP+ from NADPH which is then replaced with NADPH upon membrane depolarisation. To support this, they find PUFA containing lipids are depleted in sleep-deprived drosophila brains and based on this presume this may be due to peroxidation. They then find that deletion of the short-chain dehydrogenase sniffer which reduces aldehydes increases sleep. They go on to utilize two methods to probe the peroxidation theory, they use the miniSOG system which allows for optogenetically controlled ROS production and which they have previously demonstrated induces sleep in an hyperkinetic dependent manner presumable due to a NADPH to NADP+ switch. Secondly, they directly treat dFB neurons with the lipid peroxide product 4-ONE. With these methods they show that both interventions alter voltage-gated A-type current kinetics in a hyperkinetic manner using hyperkinetic null mutants with/without catalytically dead hyperkinetic and they show depolarisation dissipated these effects. This paper provides key evidence to build on their overall model of hyperkinetic acting as a unit to integrate increased oxidative stress/signals with a drive to sleep. However, given the central role of lipid peroxides in this study, measurement of these should be performed and increased information needs to be provided around their lipidomic analysis.

1. 4-ONE analysis.

There are no direct measurements of lipid peroxide products provided in the manuscript, yet methods to measure these via LCMS are present in the literature (eg. PMID: 23726997, PMID: 35881173) and would significantly strengthen claims around the role of these products. Specifically:

- Does 4-ONE or other lipid peroxidation products vary between rested and sleep deprived samples?
- Does 4-ONE or other lipid peroxidation products vary with the models used in figure 2?
- Do they increase with ROS induced via the miniSOG system.

2. Lipidomic analysis

The full data isn't provided for the lipidomic analysis in figure 1. In the text, its indicated that hundreds of lipids were analysed and the principal competent analysis was used to capture variance, however all that is shown in the figure is the 47 significantly different lipids. Please provide the full dataset used for statistical analysis as a supplement and the figures for any analysis referred to in the text. For example, the text currently states that in sleep deprivation, the membranes are depleted in PUFAs, however without seeing the full dataset its not possible to assess how many PUFA containing species were included in the analysis and what percentage of them are altered. Indeed features such as principal component analysis and volcano plots would help give the reader an indication of the percent of lipids that were differentially abundant and the composition of the general lipidome detected in these samples.

Referee #2

(Remarks to the Author)

This paper presents a synthesis of a chain of mechanistic links from the Hk beta subunit functioning in the Sh IA channel to the regulation of sleep via binding to long-chain polyunsaturated fatty acid peroxidation products. This is a significant new proposal that aims at uncovering the biochemical and biophysical bases of the cellular mechanisms underlying sleep

regulation. However, as summarized in the conclusion of the manuscript, the authors recognize several weaker links in the proposed chain of events that must be strengthened in order to consolidate the proposed mechanistic scheme.

Ensemble lipidomic data were collected from whole brains with and without sleep deprivation to infer conditions in a particular cell group (dFB neurons). Electrophysiological observations have been collected strictly from this single cell type and all manipulations and recordings have been performed on the soma only.

This study could have adopted a more balanced and comprehensive approach to examine closely some of the critical links by including or eliminating other plausible alternatives or certain potential interactions with additional players. It is important to project a broadened perspective, not to favor only a restricted set of players in the scenario.

Major Comments

I. As the authors pointed out in the Conclusion, the Hk beta (Kv1 beta) - Sh (Kv1) excitability control mechanism is rather pervasive among a variety of excitable cells. How would this general mechanism endow the dFB neurons' special role in sleep pressure homeostatic control? It should be further recognized that the control of Sh channel inactivation properties is not limited to Hk beta. It is well-known that the other Kv1 auxiliary subunit, qvr, has profound effect on Sh IA inactivation properties, has been implicated to interact with sod1, and can lead to extremely "sleepless" phenotypes. Furthermore, additional K channels that are sensitive to redox conditions, such as slo BK channels, could in principle take part in determining the cell firing rate and pattern. None of these are thoroughly discussed in the manuscript and they should have been examined in the same cells under the same conditions, because the mutant alleles and pharmacological tools are available.

II. dFB cells and Sh channel Hk-Beta subunit. It is surprising that the reported experimental approaches strictly focus on a single type of channels in a single type of neurons in this paper. It is important to observe how sleep deprivation and lipid peroxidation products affect cell types other than dFB cells. What if some other excitable cells have exactly the same excitability modifications just as that observed in the dFB cells? It is also surprising that Sh mutants were not examined as a control or comparison in the same study, even though clearly the Sh channel is the ultimate target proposed for the dFB neuron excitability control. Sh null mutants could serve as a good negative control since none of the manipulations described here should have any effect if the channel mediating IA is absent.

III. V-Clamp of IA & AP Firing. The descriptions of the dFB neuron firing rate and the voltage-clamp study for the Sh IA inactivation seem incomplete. First, the authors need to also characterize the recovery from inactivation kinetics and V1/2 of the activation and steady-state inactivation curves, which could be equally potent or even stronger than the inactivation kinetics (time constants) in regulating membrane excitability.

IV. It is known that interplay between Sh and other K channels can generate different firing patterns and rates. The membrane excitability patterns eventually reflects the joint actions of different K⁺ channels. The contribution from Sh IA is well recognized, but the authors need to demonstrate that the other channel types' contributions are negligible.

In the current clamp experiments, the authors emphasize the spike firing frequency only. In fact, it is well known that Sh IA is most effective in the control of time-delay to the first action potential initiation. For the repetitive firing action potential patterns, Kv2 (Shab) and slo (BK) are known to play important roles in determining the firing pattern and rate in *Drosophila* neurons. The involvement of additional channel types in dFB neuron firing control seems very likely since in sleep-deprived or ROS-stressed cells, the inactivation kinetics of IA are slowed, which should further delay the initiation of action potentials and reduced the number of action potentials in the spike train. This is the opposite to what was observed (Figure 5 shows depolarization to 75 mV from the resting potential in the soma). Some further investigation is required to resolve this issue.

V. One of the major findings reported here is that neuronal spiking activity or equivalently membrane depolarization by current injection can transiently reverse the effects of 4-ONE or local photogeneration of reactive oxygen species (using Gal4-driven miniSOG) on IA inactivation kinetics. The authors have directly demonstrated this by somatic voltage clamping to determine the local IA kinetics (Fig 6). This is an important mechanism and should be examined in some other non-dFB excitable cells as well, to see whether it is a wide-spread general mechanism. If so, it may have significant implications for the conceptual framework of sleep and restoration of the neuronal oxidative stress.

Minor Comments

I. The analogy of DRAM cited in the first paragraph on page 4 can be simplified because it does not help understanding. The current model only depends on the cumulative ensemble state (rather than spatial or temporal patterns) of Hk-beta of Sh channels in determining the whole cell excitability. Perhaps some simple hydraulic models with water level (potential) control of reservoir volume discharge would be sufficient.

II. The authors need to discuss other potential actions of PUFA peroxidation products, which are likely hydrophobic in nature. How much of their effects are mediated by Hk (and potentially via other non-Sh channels) and how much is a direct effect on membrane fluidity, which has been shown to exert great effects on the operation of different ion channels in the membrane? For examples, the conductance control by "force-from-lipid" in bacterial stretch receptor channels (studies by Martinac and Kung) and long-chain fatty acids effects on fly photoreceptor light response trp channels (papers by R Hardie).

III. It is a valuable observation that sleep deprivation leads to increased lipid peroxidation, and depletion of PUFAs

(polyunsaturated fatty acids) in the brain. However, it is even more relevant to present evidence for an increase in the endogenous peroxidation product, such as 4-ONE related compounds, in dFB cells.

IV. The voltage dependence of the Hk beta subunit binding with NADPH vs NADP⁺, a crucial step in the reversible control mechanism described above, need to be treated more thoroughly, and directly verified if possible. Otherwise, any previous biophysical studies on similar or related subjects about membrane depolarization effects on Hk beta binding properties need to be cited or discussed in detail.

V. The site of voltage clamping and current injection is the soma, which is electrically distant from the most relevant excitability control compartment, the action potential generation site. In the current-clamp experiments, the authors recognize that the recording site in the soma is electrically distant from the site of action potential generation. In voltage-clamp experiments, they applied -80 to +10 mV depolarization pulses to counteract the ROS effect on local somatic Sh IA (Fig 6). Presumably, this -80 to +10 mV pulse mimics the range of voltage swing during the action potential in the initial segment. However, this assumption needs to be explicitly stated since there may be differences in ion channel compositions and densities between the soma and other neuronal compartments.

Referee #3

(Remarks to the Author)

This is a very interesting and extremely well-argued paper. It builds upon very beautiful work from the same lab, adding some additional details to the story vis-à-vis the lipid link to AOX. The bottom line though is that this is really quite incremental- the latest installment in the evolving story, but not a new basic finding. It is important work that should be published in a specialty journal.

I also have a few problems with the conclusions- basically the links are not as tight as they should be for the claims being made.

1) causality: The authors have published work showing the influence of Sh and HK in regulating sleep, and continued in this paper to show that PUFA-derived carbonyls alter sleep and IA's inactivation time constant. To confirm that PUFA-derived carbonyls effect on sleep via this pathway, they need to directly show the effects on IA. They need to show that the catalytically dead (and the constitutively active, if exists) HK makes sleep resistant to the genetic manipulations and AOX.

2) specificity: Both ROS production and the Sh channel are global players, yet the effect on sleep is specific to certain neurons. It remains really unclear how these players influence the activity of different neurons- indeed if the model is true, all neurons which have Sh/HK should be switched on/off with sleep state. Note that in the extended figure 1, locomotion is altered as well. I think in order to claim that this is a sleep-regulation mechanism, the authors need to do the same electrophysiology experiments in cells other than dFSB neurons. And if only dFSB neurons are affected they will need to figure out why other Sh/HK containing neurons are NOT.

3) timing: If I try to simplify the mechanisms as per my understanding of the paper, it will be as follows: oxidative stress oxidizes HK's NAPH to NAP⁺. NADP⁺ prolongs IA inactivation time constant which allows dFSB neurons to be more active (hence forming the biochemical memory and encoding sleep pressure). Spiking of dFSB neurons flips NAP⁺ again to NAPH (erasing the biochemical memory and removing sleep pressure). I can be wrong but I find it difficult to get the timing of this ON-OFF mechanism to match the timing of sleep pressure. I am assuming the oxidative stress accumulates through the day until it reaches a certain threshold that turns the toggle to ON. The results show that 20 mins of simulated activity completely reverses the toggle back to OFF. The only way I can rationalize that this 20 minutes reversal underlies a sleep pressure that needs hours to reverse is to assume that there is a pool of oxidative stress products that keep toggling the switch back to ON and it takes hours to deplete them. If this is true, it needs to be demonstrated. Perhaps applying the depolarization trains (the simulated spiking) to sleep-deprived flies (without inducing more ROS production) will show different effects on the dynamics of IA inactivation time constant than in control flies?

Referee #4

(Remarks to the Author)

This manuscript proposes that a "biochemical memory redox" is at the core of the sleep homeostatic. The authors expanded previous findings by Dr. Miesenböck and colleagues (Nature 536, 333–337 (2016). <https://doi.org/10.1038/nature19055>, Nature 568, 230–234 (2019). <https://doi.org/10.1038/s41586-019-1034-5>). The merit of this work is to piece together the puzzle between reactive oxygen species (ROS), the KVβ subunit (Hyperkinetic), the voltage-gated Shaker K⁺ channel, lipid oxidation and their interplay in sleep homeostatic of Drosophila, studying the dorsal fan-shaped body neuron in the central brain. Sleep is controlled by the electrical activity of these neurons, and they are modulated by the Hyperkinetic-Shaker complex as well as by the two pore K channel (K2P "Sandman"). The authors used lipidomic analysis to show that the lipidic composition of the brain is different between well-rested and sleep deprived flies. Through genetic manipulation they showed that lipid peroxidation products increase the pressure to sleep. Combining genetic manipulation and whole-cell patch clamp technique they determined that lipid-derived carbonyls (4-oxo-2-nonenal) can affect the oxidative state of Hyperkinetic as well as ROS induced by illumination of blue light on dFB neurons expressing R23E10-GAL4-driven

miniSOG using the time constant of inactivation of Shaker as a readout. As a proposed memory biochemical mechanism, the bits are represented by the oxidative state of the NADPH-NADP⁺ bound to the Hyperkinetic. The idea that redox reaction affects the function of Hyperkinetic/Shaker or other members of the Kv1 superfamily/Kvβ is not new (Nature 369, 289–294 (1994). <https://doi.org/10.1038/369289a0>), however the proposition that they behave as memory system controlling sleep is original. Moreover, it provided functional relevance to an otherwise overlooked Kvβ subunit. The field of sleep homeostatic, biochemistry, and biophysics will benefit directly through the step forward in understanding the molecular mechanism of sleep as well as by inspiring scientists to explore the complex of the KVβ/ion channels in other tissues and their physiological relevance. Although the work presents advancements, there are major concerns that need to be addressed and they are described below.

Major concerns:

Analogy with DRAM system.

The analogy of the oxidative states of NADPH-NADP⁺ bound to Hyperkinetics to the DRAM systems is original and elegant, however, it lacks clarity. For example, in the classical conception of a DRAM, the information is stored or retrieved by a central processor unit (CPU) therefore the CPU controls what information goes to the DRAM in the format of bits. By using this analogy, although they did not explicitly propose this, the authors give the impression that the cell has a “entity” like a CPU that is controlling this proposed biochemical memory system. Moreover, it was hard to understand the proposed idea of a memory without a visual illustration. For example, on Page 3, the sentence “Imagine that tight binding of electron donor causes the oxidoreductase cycle to pause at the cofactor exchange” is too technical which makes harder for a broader audience to grasp what the authors are referring to. Therefore, an incorporation of a figure that summarizes the authors’ data/idea with the proposed mechanism of the biochemical memory and rephrasing some sentences that are too technical will improve and make the manuscript more suitable for a broad audience. Respectfully, I would suggest that the authors could use the Figure 2a shown in previous work done by Miesenböck and colleagues (Nature 568, 230–234 (2019) <https://doi.org/10.1038/s41586-019-1034-5>) as an inspiration to build such a figure.

Disagreement between the voltage-clamp and current-clamp data

The authors did not see effects of the 4-ONE on the action potential firing pattern (Figure 5) although they showed that the inactivation time constant is augmented (Figure 3 and 4). They offered an explanation stating that it was more a technical issue rather than 4-ONE altering the firing pattern independently of Kvβ (Page 8 last paragraph). This is a critical issue and must be addressed properly. Therefore, it raises the question: if such a technical issue is presented in the current-clamp experiments, why do the authors think that this is not an issue with the voltage-clamp measurements? Since they are patching the same cells, a technical issue would be present in both measurements and therefore jeopardizing their voltage-clamp/current-clamp data interpretation. This point raises the question whether the effects observed are more due to the ROS effects on PUFA rather than fragments generated by lipid oxidation (4-ONE, for example). This could be addressed by an experiment where one can use AOX and 4-ONE in the recording pipette, therefore capping ROS and making 4-ONE the only source of lipid peroxidation carbonyls as the substrate for the hyperkinetic enzymatic activity. In this context, why did the author focus exclusively on the enzymatic activity of hyperkinetic/shaker and did not consider the effects of PUFA on other voltage-gated ion channels? (Front Physiol. 2017 Feb 6;8:43. doi: 10.3389/fphys.2017.00043). Since PUFA modulates the activities of many ion channels, including K2Ps, the degradation of the PUFA would affect the function of ion channels expressed in the dFB neurons, including Sandman and Shaker. Thus, the authors must experimentally address the disagreements between the voltage-clamp and current-clamp data.

Access resistance (or series resistance) should be provided.

The authors stated in the methods section that, when the access resistance was larger than 50 MΩ (First paragraph page 17) the cells were discarded, and the pipette resistance was between 9-11 MΩ (last sentence on page 16). Therefore, the access resistance in the experiments performed was between 10 and 50 MΩ. Under voltage-clamp, the membrane capacitance (C_m) and resistance (R_m) are in parallel, and both are in series with the access resistance (R_a). Based on Figures 3, 4, 6 and Extended figure 3, the average IA current is around 500pA. Therefore, this would impose a voltage drop (VA= IA*R_a) due to R_a, ranging between 5mV (best case-scenario – R_a = 10MΩ) and 25mV (worst case scenario – 50 MΩ). The voltage drop due to the access resistance will provide serious discrepancies between the imposed voltage by the voltage command and the actual voltage applied in the cell membrane. For example, assuming the worst-case scenario, the voltage drop due to access resistance is 25mV and the voltage step used was from -110mV to +30mV (extended figure 2). The voltage that will be sensed in the membrane will be +5mV, instead of the +30mV imposed by the voltage command. This will also change the time course of the current because the error will change with the magnitude of the current over time. In this context, the authors must state whether they used a series resistance compensation and the amount used. More importantly, they should show a plot of the variability of the access resistance in the voltage-clamp experiments and discuss if these values could jeopardize their measurements.

Exemplary current trace in the presence of 4-ONE and sogMini experiments.

Although the authors have provided detailed statistical analysis of the currents recorded, the authors should also include in the figure 3, 4 and 6 exemplary current traces showing IA before and after the several treatments therein.

Minor concerns:

The figures regarding the voltage-clamp experiments are not easy to follow.

The bars indicating that the cells were held at -80mV on the top panels of Figure 3, 4, 6 and extended data 3 is confusing. It is not clear what was the test pulse used to elicit the current and assess the membrane resistance, current amplitude and time constants. This gives the impression that there was no voltage change during the recording, which might mislead the readers.

Shaker is a member of the Kv1 superfamily, but it is not the Kv1. The authors should rephrase the sentence when Shaker is introduced. Page 2: “A hint as a possible answer has come from studies in, where both the Kv1 channel Shaker and its β-subunit Hyperkinetic”.

From the figures presented in 3, 4 and 6, it is not obvious that the authors are only recording IA. They should state in the text and also refer to Extended Figure 2 where they showed the procedure to isolate the IA currents from the other ionic components.

Throughout the voltage-clamp experiment the authors held the cell membrane at -80mV. Since, holding at -80mV in presence of 4-ONE or ROS allows the Kv β to load with NADP⁺ do the authors think that if membrane were held at +30mV would impede the Kv β to load with NADP⁺?

Version 1:

Reviewer comments:

Referee #1

(Remarks to the Author)

The authors have fully addressed my issues around the lipidomic data and the 4-ONE analysis.

Referee #2

(Remarks to the Author)

The authors have made genuine efforts to revise the manuscript in response to the reviewers' comments, and the manuscript has been greatly improved. There is still one major issue remaining that can be resolved straightforwardly with data collected from a few additional cell types.

The authors state that "Our model of homeostatic sleep regulation neither states nor demands that Sh-Hk is the only determinant of dFBN activity, nor that the mechanism of excitability control via Hk is exclusive to dFBNs. Our experiments concentrate on a particular ion channel in a specific type of neuron in order to test a sharply stated hypothesis: that Hk forms a voltage-gated lipid peroxidation memory whose contents in dFBNs influence sleep." They resisted the suggestion to investigate the same physiological responses in non-dFBN neurons.

As the authors suggest, the Sh-Hk complex used as a lipid peroxidation sensor to convert the oxidative stress levels to neuronal excitability could be presumed as a common mechanism shared by many neurons. However, there is no guarantee that no additional factors or modified molecular properties specific to the dFBNs are involved to facilitate or tune the Sh-Hk complex sensor properties. There is no shortage of neuronal types in the fly brain that can be analyzed with the same physiologically experiments reported here. In fact, there are larger neurons that can be used for quick examinations.

It is indeed true that, as the authors pointed out in the analogy, all neurons use the common currency of action potentials to perform endless types of tasks. However, the essence is that different wave forms and firing patterns characteristic of different neuronal types are seen in the nervous system. To gain a deeper understanding of nervous system function, the action potential wave form and firing patterns should not be ignored.

Whereas the reader could agree with the proposal that the Sh-Hk complex is involved in the process, the natural question is whether it works the same way in dFBNs as in other neurons which all have the Sh-Hk complex in excitability control. To make this paper more interesting to a broader readership of the journal, the small investment in collecting data from some other neuronal types to resolve this issue could enhance the impact of this study.

Referee #4

(Remarks to the Author)

The authors have significantly improved the text and the revised version was easier to follow, but there are still places that need further clarification. I recommend the incorporation of figure B (shown in the rebuttal) since it depicts in a simple manner the DRAM analogy and the proposed mechanism. There is no question about the quality of the work and the care that the researchers took in conducting the experiments. While the manuscript presents compelling evidence supporting their hypothesis, the discrepancies between the current-clamp and voltage-clamp data undermine the robustness of the conclusions drawn. The voltage-clamp data is extensively tested and using different controls, which is lacking in the current-clamp data. In my opinion, I respectfully argue that the current-clamp data is the most important evidence for their hypothesis since it reflects more closely the physiological condition and response of the dFBNs. I do understand the experimental challenges that the authors are dealing with when performing electrophysiological assays in the dFBNs. The space-clamp issue that they reported as one of the possible explanations for the lack of effects in the 4-ONE experiments will be present in both current and voltage-clamp experiments. While it is appreciated that the authors acknowledged difficulties and tried to provide explanation for their technical issues, I disagree with their explanation. Thus, I recommend that they address the following concerns regarding their explanation for the current-clamp data which, in my opinion, was not addressed by the authors in the revised form of the manuscript and will increase the robustness of their conclusions.

1) The current-clamp and voltage-clamp disagreement data still needs clarification. The fact the 4-ONE is still active in the recording for the voltage clamp experiments even after 40 minutes of the experiments (Fig. 6d), does not align with the idea of half-life in tissue (<4s) as the cause of the lack of effect of 4-ONE in the spike response shown current-clamp experiments (Fig 5b). Thus, I respectfully disagree about the short half-life of the 4-ONE as the reason for the current-clamp data technical issue postulated in the page 9.

2) The authors also justify the lack of increase in the spike response due to the low capability of the diffusion of the 4-ONE to the dendrites and axons, which does not happen in the miniSOG scenario. Thus, one can conclude that the dendrites and axons can be more effective in spike initiation in the dFBNs than the soma does. In order to demonstrate the lack of the

effects of the 4-ONE due to the poor diffusion to the dendrites and axons, the authors can perform control experiments where they use miniSOG. Since miniSOG would be present in the soma, dendrites, and axons they could illuminate different regions to support their interpretation regarding the poor diffusion of 4-ONE. 1) They can illuminate the dendrites and axons regions only. This will test the hypothesis that those regions are more effective than the soma to increase the spike response. 2) They can only illuminate the soma when miniSOG is present. In this condition only Shaker present at the soma will be affected by ROS, and according to their explanation for poor diffusion of 4-ONE, this will not lead to an increase in the spike response. 3) They must run a control where they illuminate the dFBNs only with LED without the expression of miniSOG to fully demonstrate that the LED by itself is not leading to the increase in the spike response by unintended effects such as increase in temperature, for example. This needs to be done using the same illumination condition that they tested in the data reported in this study.

Figure 5 c-e: it is hard to understand what the authors are trying to show in the figure. Are they trying to compare different expression levels of Hyperkinetic or just to show how the soma, dendrites and axons are distributed? To improve clarity, they can label in the figure the soma, the dendrites, and axons so the paper can be more appreciated by readers not familiar with the anatomy of the drosophila brain.

Minor comments:

Extended Figure 6: what are the conditions of the traces (gray and the turquoise, and yellow and turquoise, respectively) in b and d? Do they see the same rundown effects as in the IA? It is appreciated that the authors did controls using the mouse Kv1.4 and Kv β 2 subunit, however, I wonder why the Shaker and the Hyperkinetic for control weren't used, since they would be the ideal system in this case.

Version 2:

Reviewer comments:

Referee #2

(Remarks to the Author)

In this revised manuscript, the authors have satisfactorily addressed the issues raised in my previous review.

Referee #4

(Remarks to the Author)

The authors have addressed my concerns. They added new experimental data in the manuscript that strength their hypothesis. I fully support the publication.

Response to Reviewers of Nature Manuscript 2023-04-05806

We thank the referees for their comments and suggestions.

Referee 1

In this study, the authors build on previous investigations proposing that the beta subunits of the Kv1 channels in drosophila(hyperkinetic) are important for sleep regulation via the status of its NADP⁺ cofactor which varies in response to mitochondrial ROS. Here they extend this model to propose that lipid peroxidation products, primarily 4-ONE mediate this communication and cause formation of NADP⁺ from NADPH which is then replaced with NADPH upon membrane depolarisation. To support this, they find PUFA containing lipids are depleted in sleep-deprived drosophila brains and based on this presume this may be due to peroxidation. They then find that deletion of the short-chain dehydrogenase sniffer which reduces aldehydes increases sleep. They go on to utilize two methods to probe the peroxidation theory, they use the miniSOG system which allows for optogenetically controlled ROS production and which they have previously demonstrated induces sleep in an hyperkinetic dependent manner presumable due to a NADPH to NADP⁺ switch. Secondly, they directly treat dFB neurons with the lipid peroxide product 4-ONE. With these methods they show that both interventions alter voltage-gated A-type current kinetics in a hyperkinetic manner using hyperkinetic null mutants with/without catalytically dead hyperkinetic and they show depolarisation dissipated these effects. This paper provides key evidence to build on their overall model of hyperkinetic acting as a unit to integrate increased oxidative stress/signals with a drive to sleep. However, given the central role of lipid peroxides in this study, measurement of these should be performed and increased information needs to be provided around their lipidomic analysis.

1. 4-ONE analysis.

There are no direct measurements of lipid peroxide products provided in the manuscript, yet methods to measure these via LCMS are present in the literature (eg. PMID: 23726997, PMID: 35881173) and would significantly strengthen claims around the role of these products. Specifically:

- Does 4-ONE or other lipid peroxidation products vary between rested and sleep deprived samples?
- Does 4-ONE or other lipid peroxidation products vary with the models used in figure 2?
- Do they increase with ROS induced via the miniSOG system.

The amount of material required for LC-MS² detection of 4-ONE and other PUFA-derived carbonyls is prohibitive for studies in Drosophila. The reference cited by the reviewer uses 100 mg of tissue¹; for comparison, the LC-MS² validation of our SMALDI-MSI hits was done on 9 mg of brain tissue, which was harvested over a period of three months in daily batches totalling 1,300 individually sleep-deprived and hand-dissected flies (plus another 1,300 hand-dissected control flies). It would be impossible to perform such an experiment at a ten-fold larger scale.

We therefore concentrated on the approach described in the second cited reference², namely SMALDI-MSI, but found our efforts hampered by the short half-life of reactive electrophiles in biological samples^{3,4}. Spotting a drop of 100 μ M 4-ONE standard onto a clean glass slide gave rise to a large signal at the expected mass following derivatization

with Girard's reagent T, but the signal became undetectable if the same amount of 4-ONE was spiked onto a cryosection of a fly brain (Extended Data Fig. 7a). We attribute the loss of signal to the rapid formation of covalent adducts between 4-ONE and endogenous nucleophiles, consistent with a reported 4-ONE half-life of <4 s in tissue³. Trace amounts of 4-ONE detected in rested but not sleep-deprived brains cannot correspond to endogenous levels of 4-ONE at the moment of dissection, which would have been vastly elevated in *sni¹* mutants but were no longer at the time of measurement (Extended Data Fig. 7b). They must instead reflect oxidation products formed during the 2-h incubation with Girard's reagent, which can capture nascent 4-ONE by out-competing endogenous nucleophiles. Because the PUFA content of membranes in rested brains is higher (Fig. 1d), so is their oxidizability during analysis. We present these results and their interpretation in the new Extended Data Fig. 7 and the discussion section of the manuscript.

2. Lipidomic analysis

The full data isn't provided for the lipidomic analysis in figure 1. In the text, it is indicated that hundreds of lipids were analysed and the principal component analysis was used to capture variance, however all that is shown in the figure is the 47 significantly different lipids. Please provide the full dataset used for statistical analysis as a supplement and the figures for any analysis referred to in the text. For example, the text currently states that in sleep deprivation, the membranes are depleted in PUFAs, however without seeing the full dataset it is not possible to assess how many PUFA containing species were included in the analysis and what percentage of them are altered. Indeed features such as principal component analysis and volcano plots would help give the reader an indication of the percent of lipids that were differentially abundant and the composition of the general lipidome detected in these samples.

The full SMALDI-MSI dataset has been uploaded to metaspace (hosted by EMBL); the full LC-MS² dataset has been uploaded to MassIVE Repository (hosted by UCSD); source data for Fig. 1 are included in the Source Data Excel file. The datasets are currently embargoed, but we would be happy to arrange access for peer review via Nature's editorial office.

We thank the reviewer for prompting us to analyse our lipidomic data more fully. The revised Fig. 1c presents a volcano plot of all 380 signals annotated as glycerophospholipids; Fig. 1d contains the results of statistical enrichment analyses of the effect of sleep history on the number of double bond equivalents and different types of head group.

Referee 2

This paper presents a synthesis of a chain of mechanistic links from the Hk beta subunit functioning in the Sh I_A channel to the regulation of sleep via binding to long-chain polyunsaturated fatty acid peroxidation products. This is a significant new proposal that aims at uncovering the biochemical and biophysical bases of the cellular mechanisms underlying sleep regulation. However, as summarized in the conclusion of the manuscript, the authors recognize several weaker links in the proposed chain of events that must be strengthened in order to consolidate the proposed mechanistic scheme.

Ensemble lipidomic data were collected from whole brains with and without sleep deprivation to infer conditions in a particular cell group (dFB neurons). Electrophysiological observations have been collected strictly from this single cell type and all manipulations and recordings have been performed on the soma only.

The aim of our SMALDI-MSI analysis was to detect spatially resolved lipidomic changes across the brain, not to infer conditions in dFBNs from whole-brain data. We detail in our response to Referee 3 how we think metabolic conditions in dFBNs relate to those in other neurons and support these ideas with new experimental evidence in a companion manuscript⁵.

*The small size of central neurons in *Drosophila* precludes electrophysiological measurements from dendritic or axonal processes; imaging techniques cannot resolve the inactivation kinetics of I_A , the experimental variable of interest; and we can think of neither methodology nor rationale for non-somatic genetic or pharmacological manipulations.*

We ask the reviewer to bear in mind that what may appear routine in two-electrode voltage-clamp measurements on larval muscle is at the very limit of what is technically possible in whole-cell patch-clamp recordings from central neurons in the adult, especially as our voltage clamp is required to last for 30 minutes or more.

This study could have adopted a more balanced and comprehensive approach to examine closely some of the critical links by including or eliminating other plausible alternatives or certain potential interactions with additional players. It is important to project a broadened perspective, not to favor only a restricted set of players in the scenario.

We thank Referees 2 and 3 for highlighting a possible source of misunderstanding, which we now clarify in the concluding paragraph of the manuscript. Our model of homeostatic sleep regulation neither states nor demands that Sh-Hk is the only determinant of dFBN activity, nor that the mechanism of excitability control via Hk is exclusive to dFBNs. Our experiments concentrate on a particular ion channel in a specific type of neuron in order to test a sharply stated hypothesis: that Hk forms a voltage-gated lipid peroxidation memory whose contents in dFBNs influence sleep.

*We have taken care not to create the impression that there are no other influences on the activity of dFBNs; in fact, we ourselves have identified some of them in earlier work^{6,7}. However, these factors lie outside the scope of the present study. The introduction to our manuscript makes explicit that ‘the electrical activity of [dFBNs] fluctuates—in part—because Hyperkinetic modulates the inactivation kinetics of the Shaker current.’ Specific comments on two potential players proposed for further investigation by the reviewer, quiver (*qvr*) and slowpoke (*slo*), and on insights that might be gleaned from examining other neuron types electrophysiologically, appear below.*

Major Comments

I. As the authors pointed out in the Conclusion, the Hk beta (Kv1 beta) - Sh (Kv1) excitability control mechanism is rather pervasive among a variety of excitable cells. How would this general mechanism endow the dFB neurons’ special role in sleep pressure homeostatic control? It should be further recognized that the control of Sh channel inactivation properties is not limited to Hk beta. It is well-known that the other Kv1 auxiliary subunit, *qvr*, has profound effect on Sh I_A inactivation properties, has been implicated to interact with *sod1*, and can lead to extremely “sleepless” phenotypes. Furthermore, additional K channels that are sensitive to redox conditions, such as *slo* BK channels, could in principle take part in determining the cell firing rate and pattern. None of these are thoroughly discussed in the manuscript and they should have been examined in the same cells under the same conditions, because the mutant alleles and pharmacological tools are available.

A possibly universal role of potassium channel β -subunits in coupling mitochondrial respiration and/or lipid peroxidation to neuronal excitability does not preclude that dFBNs harness this general mechanism for the special purpose of regulating sleep. By analogy, the action potential is a universal mechanism for neuronal communication, but different neuron types use this mechanism to different ends: a retinal ganglion cell spike means something different from the spike of a striatal medium spiny neuron, even though both are generated by the same type of voltage-gated sodium channel. The blindness that follows the elimination of sodium channels from retinal ganglion cells (but not medium spiny neurons) justifies the conclusion that these cells and their sodium spikes are important for vision; whether or not other neurons also use sodium channels to generate action potentials has no bearing on the validity of this conclusion. By the same token, we consider excitability control via Hk a likely general mechanism that is used by dFBNs for the special purpose of regulating sleep. What sets dFBNs apart from other neurons is not the absence or presence of Hk but their anti-cyclical energy metabolism, as we demonstrate in a companion manuscript⁵ and summarize in our response to Referee 3.

*While we do not question that quiver (*qvr*) can influence the inactivation kinetics of Shaker, there is no biochemical, biophysical, or structural evidence for reversible, redox-dependent modulation of the channel through *qvr*. An interaction between *qvr* and superoxide dismutase 1 (*SOD1*) has been described at the population level, where the presence of *qvr* mutant 'helper' flies extends the lifespan of short-lived *SOD1* mutants⁸. The effect does not depend on or imply a physical interaction between the two proteins or indicate that *qvr* is redox-sensitive; it is not specific to *qvr* mutant flies but also seen with other genotypes; and it appears to reflect the instigation of motor activity by helpers rather than the transmission of a redox signal.*

*The revised manuscript includes a new demonstration that the carbonyl- and voltage-driven changes in the inactivation kinetics of the Sh-Hk complex in flies are recapitulated in HEK-293 cells expressing mouse *Kv1.4* and *Kv β 2* (Extended Data Fig. 6). This reduced system provides a compelling argument against the involvement of other auxiliary subunits in the response to PUFA-derived carbonyls.*

**Drosophila* slowpoke (*slo*) lacks conserved cysteines that underlie the redox sensitivity of the human channel and is therefore unaffected by changes in redox potential⁹. consistent with the reported lack of a sleep phenotype in *slo* mutants¹⁰.*

[REDACTED]

II. dFB cells and Sh channel Hk-Beta subunit. It is surprising that the reported experimental approaches strictly focus on a single type of channels in a single type of neurons in this paper. It is important to observe how sleep deprivation and lipid peroxidation products affect cell types other than dFB cells. What if some other excitable cells have exactly the same excitability modifications just as that observed in the dFB cells? It is also surprising that Sh mutants were not examined as a control or comparison in the same study, even though clearly the Sh channel is the ultimate target proposed for the dFB neuron excitability control. Sh null

mutants could serve as a good negative control since none of the manipulations described here should have any effect if the channel mediating I_A is absent.

*Our strict focus on Hk and dFBNs is based on prior work, which established a central sleep-regulatory role for $K_V\beta$ in these neurons^{7,11}. Our revisions strengthen this case with a demonstration that the RNAi-mediated depletion of Hk only from dFBNs corrects the excessive sleep of *sni¹* mutants at least as effectively as the pan-neuronal knockdown of Hk does (Fig. 2b, c). This restricts the sleep-regulatory function of Hk to dFBNs, irrespective of whether or not other excitable cells also exhibit Hk-dependent biophysical changes.*

Sh was established as the ultimate target of redox-mediated excitability changes in an earlier study, which showed that the absence of Sh or Hk from dFBNs occludes the sleep-promoting effect of pro-oxidant manipulations¹¹. These manipulations included the expression of a pro-oxidant version of SOD1 and the light-driven generation of singlet oxygen by miniSOG.

Shaker null mutants lack I_A and therefore would not allow measurements of the current's inactivation kinetics, the variable of interest in our study.

III. V-Clamp of I_A & AP Firing. The descriptions of the dFB neuron firing rate and the voltage-clamp study for the Sh I_A inactivation seem incomplete. First, the authors need to also characterize the recovery from inactivation kinetics and $V_{1/2}$ of the activation and steady-state inactivation curves, which could be equally potent or even stronger than the inactivation kinetics (time constants) in regulating membrane excitability.

The new Extended Data Fig. 4c reports steady-state activation and inactivation curves.

IV. It is known that interplay between Sh and other K channels can generate different firing patterns and rates. The membrane excitability patterns eventually reflects the joint actions of different K^+ channels. The contribution from Sh I_A is well recognized, but the authors need to demonstrate that the other channel types' contributions are negligible.

In the current clamp experiments, the authors emphasize the spike firing frequency only. In fact, it is well known that Sh I_A is most effective in the control of time-delay to the first action potential initiation. For the repetitive firing action potential patterns, $Kv2$ (Shab) and slo (BK) are known to play important roles in determining the firing pattern and rate in *Drosophila* neurons. The involvement of additional channel types in dFB neuron firing control seems very likely since in sleep-deprived or ROS-stressed cells, the inactivation kinetics of I_A are slowed, which should further delay the initiation of action potentials and reduced the number of action potentials in the spike train. This is the opposite to what was observed (Figure 5 shows depolarization to 75 mV from the resting potential in the soma). Some further investigation is required to resolve this issue.

We neither claim that the contributions of other channel types to the excitability patterns of dFBNs are negligible, nor does our model require that they are. Our hypothesis states that dFBNs estimate sleep pressure by monitoring the flow of electrons through their own mitochondria; that sleep loss diverts high-energy electrons from the respiratory chain into uncontrolled side reactions with O_2 ; that the resulting ROS cause PUFA chains of membrane lipids to fragment into short- or medium-chain carbonyls; and that Hk is the dFBN-intrinsic transducer that converts this biochemical signal into firing rate

changes. Unequivocal support for this hypothesis comes from direct interference at multiple levels: mitochondrial electron and proton transport, mitochondrial fission and fusion machinery, pro- and antioxidant enzymes, carbonyl reductase, and Sh and Hk^{5,7,11}. Without exception, these interventions produce the predicted increases or decreases in sleep, and these go hand-in-hand with increases or decreases in the excitability of dFBNs^{5,7,11}.

Attenuating I_A —whether by RNAi knockdown of Sh or by increasing the likelihood of inactivation via a variety of means—consistently reduces the spike frequency of dFBNs^{7,11}. A recently published Hodgkin–Huxley model based on our empirical parameters reproduces the relationship between I_A inactivation and firing rate¹².

V. One of the major findings reported here is that neuronal spiking activity or equivalently membrane depolarization by current injection can transiently reverse the effects of 4-ONE or local photogeneration of reactive oxygen species (using Gal4-driven miniSOG) on I_A inactivation kinetics. The authors have directly demonstrated this by somatic voltage clamping to determine the local I_A kinetics (Fig 6). This is an important mechanism and should be examined in some other non-dFB excitable cells as well, to see whether it is a wide-spread general mechanism. If so, it may have significant implications for the conceptual framework of sleep and restoration of the neuronal oxidative stress.

The fusion of an epitope tag to endogenous Hk has allowed us to visualize its distribution in the brain as part of our revisions (the new Fig. 5c–e). Given how widespread this distribution is (as expected, though with a notable enrichment of Hk in layers of the fan-shaped body targeted by dFBNs), the mechanism we describe is in all likelihood general and adopted by dFBNs for the special purpose of regulating sleep. As explained above, our hypothesis is indifferent to the presence or absence of Hk-mediated excitability control in non-dFBN cells.

*A mammalian K_v1 family member in a simple heterologous system—HEK-293 cells coexpressing mouse $K_v1.4$ and $K_v\beta2$ —exhibits stable carbonyl-driven increases in its inactivation time constants that are reversed by cycles of membrane depolarization (Extended Data Fig. 6). This result shows that sensitivity to PUFA-derived carbonyls is an intrinsic property of K_v1 family members that generalizes to species other than *Drosophila*.*

Minor Comments

I. The analogy of DRAM cited in the first paragraph on page 4 can be simplified because it does not help understanding. The current model only depends on the cumulative ensemble state (rather than spatial or temporal patterns) of Hk-beta of Sh channels in determining the whole cell excitability. Perhaps some simple hydraulic models with water level (potential) control of reservoir volume discharge would be sufficient.

We prefer to retain the analogy with DRAM, which often elicits a noticeable flicker of understanding in the audience during presentations of this work. The analogy refers to the two-element architecture of the elementary storage unit in Dennard's patent^{13,14}—one transistor and one capacitor, which we compare to the α - and β -subunits of the potassium channel—rather than spatial or temporal patterns in an array of such units. We have rephrased the passage to make this clear and could include a simple illustration if reviewers and editors think that this would be a valuable addition (please see Fig. B in our response to Referee 4).

II. The authors need to discuss other potential actions of PUFA peroxidation products, which are likely hydrophobic in nature. How much of their effects are mediated by Hk (and potentially via other non-Sh channels) and how much is a direct effect on membrane fluidity, which has been shown to exert great effects on the operation of different ion channels in the membrane? For examples, the conductance control by “force-from-lipid” in bacterial stretch receptor channels (studies by Martinac and Kung) and long-chain fatty acids effects on fly photoreceptor light response trp channels (papers by R Hardie).

Short- to medium chain carbonyls are thought to have enough solubility in water to dissociate from membranes⁴, allowing them to reach the active site of Hk. Extended Data Fig. 5 demonstrates unambiguously that all effects of 4-ONE on I_A are mediated by its reduction at the active site of Hk, as the substitution of a catalytically dead version for functional Hk in the intact channel renders I_A insensitive to 4-ONE (Extended Data Fig. 5a, b). This excludes effects via non-specific modifications of the channel protein or changes in membrane fluidity.

III. It is a valuable observation that sleep deprivation leads to increased lipid peroxidation, and depletion of PUFAs (polyunsaturated fatty acids) in the brain. However, it is even more relevant to present evidence for an increase in the endogenous peroxidation product, such as 4-ONE related compounds, in dFB cells.

Our attempts to obtain such measurements are summarized in our response to Referee 1, who made a similar request. The results are presented in Extended Data Fig. 7 and discussed in the penultimate paragraph of the revised manuscript.

IV. The voltage dependence of the Hk beta subunit binding with NADPH vs NADP⁺, a crucial step in the reversible control mechanism described above, need to be treated more thoroughly, and directly verified if possible. Otherwise, any previous biophysical studies on similar or related subjects about membrane depolarization effects on Hk beta binding properties need to be cited or discussed in detail.

The inactivation kinetics of I_A are the only practical way of inferring the oxidation state of the cofactor in vivo; direct verification, for example by mass spectrometry or NADPH fluorescence spectroscopy of the intact channel complex, would require the purification of impossibly large quantities of protein from a small number of neurons. Reference 8 cites the original proposal that voltage regulates cofactor binding¹⁵; reference 26 points to an experimental demonstration in oocytes that the holding potential influences the rates of hydride transfer and cofactor exchange¹⁶. The influence is modest, presumably because the membrane voltage was clamped at constant hyper- or depolarized values rather than alternated between them¹⁶.

V. The site of voltage clamping and current injection is the soma, which is electrically distant from the most relevant excitability control compartment, the action potential generation site. In the current-clamp experiments, the authors recognize that the recording site in the soma is electrically distant from the site of action potential generation. In voltage-clamp experiments, they applied -80 to +10 mV depolarization pulses to counteract the ROS effect on local somatic Sh I_A (Fig 6). Presumably, this -80 to +10 mV pulse mimics the range of voltage swing during the action potential in the initial segment. However, this assumption needs to be

explicitly stated since there may be differences in ion channel compositions and densities between the soma and other neuronal compartments.

We have included the requested cautionary statement and a reference about space-clamp problems on p. 10.

Referee 3

This is a very interesting and extremely well-argued paper. It builds upon very beautiful work from the same lab, adding some additional details to the story vis-à-vis the lipid link to AOX. The bottom line though is that this is really quite incremental- the latest installment in the evolving story, but not a new basic finding. It is important work that should be published in a specialty journal.

We thank the referee for the words of praise but respectfully disagree with the somewhat self-contradictory opinion that progress has been incremental. The study establishes lipid peroxidation as a cause of sleep; it uncovers the molecular mechanism that links events in the inner mitochondrial membrane to changes in neuronal excitability; it assigns a function to the mysterious potassium channel β -subunit; and it introduces the idea of a voltage-gated enzymatic cycle as a device for memory storage. Each of these conceptual advances is significant.

I also have a few problems with the conclusions- basically the links are not as tight as they should be for the claims being made.

1) causality: The authors have published work showing the influence of Sh and HK in regulating sleep, and continued in this paper to show that PUFA-derived carbonyls alter sleep and I_A 's inactivation time constant. To confirm that PUFA-derived carbonyls effect on sleep via this pathway, they need to directly show the effects on I_A . They need to show that the catalytically dead (and the constitutively active, if exists) HK makes sleep resistant to the genetic manipulations and AOX.

Our revisions include a demonstration that the RNAi-mediated depletion of Hk from dFBNs makes sleep resistant to the high carbonyl load of sni^1 mutants (Fig. 2b, c). This experiment is equivalent to (but more practical to realize than) the use of catalytically dead Hk, which would require the recombination of two X-linked mutant alleles (Hk^1 and sni^1) plus the expression of the Hk^{K289M} transgene. Isolating Hk^1 recombinants is problematic because the nature of the molecular lesion is unknown.

Hk-deficient dFBNs fail to promote sleep upon illumination of miniSOG, as shown previously¹¹. Constitutively active versions of Hk do, unfortunately, not exist.

The expression of either catalytically dead Hk or AOX in dFBNs causes a similar degree of sleep loss, and so does their co-expression. Although this result is consistent with the proposed pathway, in which AOX acts upstream of Hk, it cannot speak directly to the involvement of PUFA-derived carbonyls and is therefore not part of the revised manuscript.

2) specificity: Both ROS production and the Sh channel are global players, yet the effect on sleep is specific to certain neurons. It remains really unclear how these players influence the activity of different neurons- indeed if the model is true, all neurons which have Sh/HK

should be switched on/off with sleep state. Note that in the extended figure 1, locomotion is altered as well. I think in order to claim that this is a sleep-regulation mechanism, the authors need to do the same electrophysiology experiments in cells other than dFSB neurons. And if only dFSB neurons are affected they will need to figure out why other Sh/HK containing neurons are NOT.

We thank Referees 2 and 3 for highlighting a potential source of misunderstanding, which we now clarify in the concluding paragraph of the manuscript. A possibly ubiquitous role of potassium channel β -subunits in coupling mitochondrial respiration and/or lipid peroxidation to neuronal excitability changes does not preclude that dFBNs harness this general mechanism for the special purpose of regulating sleep. By analogy, the action potential is a universal mechanism for neuronal communication, but different neuron types use this mechanism to different ends: a retinal ganglion cell spike means something different from the spike of a striatal medium spiny neuron, even though both are generated by the same type of voltage-gated sodium channel. The blindness that follows the elimination of sodium channels from retinal ganglion cells justifies the conclusion that these cells and their sodium spikes are important for vision; whether or not other neurons also use sodium channels to generate action potentials has no bearing on the validity of this conclusion. By the same token, we consider excitability control via Hk a likely general mechanism that is used by dFBNs for the special purpose of regulating sleep.

What sets dFBNs apart from other neurons is not the absence or presence of Hk but their anti-cyclical energy metabolism, as we show in a companion manuscript⁵. When the demands of ATP synthesis are high, the vast majority of electrons entering the mitochondrial transport chain reach O_2 in an enzymatic reaction catalyzed by cytochrome c oxidase (complex IV); only a small minority leak prematurely from the upstream mobile carrier coenzyme Q (CoQ), producing superoxide and other ROS^{17,18}. The probability of these non-enzymatic single-electron reductions of O_2 increases sharply under conditions that overflow the CoQ pool as a consequence of increased supply (high NADH/NAD⁺ ratio) or reduced demand (large protonmotive force and high ATP/ADP ratio)^{17,18}. The mitochondria of dFBNs are prone to this mode of operation during waking¹¹, when caloric intake is high but the neurons' electrical activity is reduced, leaving their ATP reserves full. Indeed, measurements with genetically encoded sensors reveal approximately 15-fold higher ATP concentrations in dFBNs, but not in olfactory projection neurons, after sleep deprivation than at rest⁵.

Even if the chance of an individual electron spilling from the CoQ pool is low, however, metabolically highly active cells, such as neurons, will by the sheer number of electrons passing through their respiratory chains generate significant amounts of ROS¹⁷⁻¹⁹. Their anti-cyclical relationship between energy availability (which peaks during waking) and energy consumption (which peaks during sleep) predisposes dFBNs to an exaggerated form of the electron leak experienced by many neurons in the awake state, making them an effective early warning system against widespread damage.

3) timing: If I try to simplify the mechanisms as per my understanding of the paper, it will be as follows: oxidative stress oxidizes HK's NAPH to NAP⁺. NADP⁺ prolongs I_A inactivation time constant which allows dFSB neurons to be more active (hence forming the biochemical memory and encoding sleep pressure). Spiking of dFSB neurons flips NAP⁺ again to NAPH (erasing the biochemical memory and removing sleep pressure). I can be wrong but I find it difficult to get the timing of this ON-OFF mechanism to match the timing of sleep pressure. I

am assuming the oxidative stress accumulates through the day until it reaches a certain threshold that turns the toggle to ON. The results show that 20 mins of simulated activity completely reverses the toggle back to OFF. The only way I can rationalize that this 20 minutes reversal underlies a sleep pressure that needs hours to reverse is to assume that there is a pool of oxidative stress products that keep toggling the switch back to ON and it takes hours to deplete them. If this is true, it needs to be demonstrated. Perhaps applying the depolarization trains (the simulated spiking) to sleep-deprived flies (without inducing more ROS production) will show different effects on the dynamics of I_A inactivation time constant than in control flies?

There are too many unknowns for a direct comparison of the natural sleep-wake cycle with the time scale at which PUFA-derived carbonyls and electrical activity alter the inactivation kinetics of I_A in our experiments. We do not know how physiological levels of PUFA-derived carbonyls compare with those of 4-ONE in our patch pipette; we do not know the rate at which miniSOG generates ROS and peroxidized lipids relative to the mitochondria of dFBNs; and we do not know how the natural spike frequencies of dFBNs compare with the intense depolarization cycles we impose. The technical difficulty of holding voltage-clamp recordings for more than a few minutes forced us to drive the channel as quickly as possible through what may be slower natural changes, by applying sufficiently high doses of 4-ONE or light and high-frequency voltage steps.

That said, several pieces of information suggest at least a rough timing match between changes in channel biophysics and sleep behaviour. First, sleep induced by a 9-minute light exposure of flies expressing miniSOG in dFBNs outlasts the illumination period by approximately one hour¹¹; sleep precipitated by powering ATP synthesis in dFBN mitochondria with a light-driven proton pump outlasts the 1-h light exposure by several hours⁵. The artificially induced, Hk-dependent sleep pressure jumps in these experiments thus take more than 20 minutes to reverse (and thereby attest to the redox memory in action).

*Second, sleep in *Drosophila* is fragmented, with an average episode duration of only 30 minutes (Extended Data Fig. 1b)—identical to the length of one cofactor oxidation and exchange cycle in our patch-clamp experiments. Loss of the carbonyl reductase *sni* extends the average sleep episode duration by 11-fold to 5.5 h (Extended Data Fig. 1b), with *sni*¹ mutants not infrequently sleeping through the night. This may be close to the scenario envisioned by the reviewer, where ‘there is a pool of oxidative stress products that keep toggling the switch back to ON and it takes hours to deplete them.’*

Referee 4

This manuscript proposes that a “biochemical memory redox” is at the core of the sleep homeostatic. The authors expanded previous findings by Dr. Miesenböck and colleagues (Nature 536, 333–337 (2016), Nature 568, 230–234 (2019)). The merit of this work is to piece together the puzzle between reactive oxygen species (ROS), the KV β subunit (Hyperkinetic), the voltage-gated Shaker K⁺ channel, lipid oxidation and their interplay in sleep homeostatic of *Drosophila*, studying the dorsal fan-shaped body neuron in the central brain. Sleep is controlled by the electrical activity of these neurons, and they are modulated by the Hyperkinetic-Shaker complex as well as by the two pore K channel (K2P “Sandman”). The authors used lipidomic analysis to show that the lipidic composition of the brain is different between well-rested and sleep deprived flies. Through genetic manipulation they showed that lipid peroxidation products increase the pressure to sleep. Combining genetic manipulation and

whole-cell patch clamp technique they determined that lipid-derived carbonyls (4-oxo-2-nonenal) can affect the oxidative state of Hyperkinetic as well as ROS induced by illumination of blue light on dFB neurons expressing R23E10-GAL4-driven miniSOG using the time constant of inactivation of Shaker as a readout. As a proposed memory biochemical mechanism, the bits are represented by the oxidative state of the NADPH-NADP⁺ bound to the Hyperkinetic. The idea that redox reaction affects the function of Hyperkinetic/Shaker or other members of the Kv1 superfamily/Kvβ is not new (Nature 369, 289–294 (1994)), however the proposition that they behave as memory system controlling sleep is original. Moreover, it provided functional relevance to an otherwise overlooked Kvβ subunit. The field of sleep homeostatic, biochemistry, and biophysics will benefit directly through the step forward in understanding the molecular mechanism of sleep as well as by inspiring scientists to explore the complex of the Kvβ/ion channels in other tissues and their physiological relevance. Although the work presents advancements, there are major concerns that need to be addressed and they are described below.

Major concerns:

Analogy with DRAM system.

The analogy of the oxidative states of NADPH-NADP⁺ bound to Hyperkinetics to the DRAM systems is original and elegant, however, it lacks clarity. For example, in the classical conception of a DRAM, the information is stored or retrieved by a central processor unit (CPU) therefore the CPU controls what information goes to the DRAM in the format of bits. By using this analogy, although they did not explicitly propose this, the authors give the impression that the cell has a “entity” like a CPU that is controlling this proposed biochemical memory system. Moreover, it was hard to understand the proposed idea of a memory without a visual illustration. For example, on Page 3, the sentence “Imagine that tight binding of electron donor causes the oxidoreductase cycle to pause at the cofactor exchange” is too technical which makes harder for a broader audience to grasp what the authors are referring to. Therefore, an incorporation of a figure that summarizes the authors’ data/idea with the proposed mechanism of the biochemical memory and rephrasing some sentences that are too technical will improve and make the manuscript more suitable for a broad audience. Respectfully, I would suggest that the authors could use the Figure 2a shown in previous work done by Miesenböck and colleagues (Nature 568, 230–234 (2019)) as an inspiration to build such a figure.

The analogy with DRAM refers to the two-element architecture of the elementary storage unit in Dennard’s patent^{13,14}—one transistor and one capacitor, which we compare to the α- and β-subunits of the potassium channel—rather than an array of many such units, controlled by a CPU. We have rephrased the passage to make this clear, edited some of the more technical language at the reviewer’s suggestion, and would be happy to include the simple illustration below if reviewers and editors think that this would be a valuable addition (Fig. B).

Fig. B | Information storage in DRAM memory cells and Kv1 channels.

a, The bits '0' (left) and '1' (centre) are represented by charges on the storage capacitor of a DRAM cell. The memory is read out when the voltage across the access transistor gate goes high and the capacitor discharges (right). **b**, The bits '0' (left) and '1' (centre) are represented by the oxidation states of the cofactor bound to the Kv β subunit of a Kv1 channel. The memory is read out when the membrane voltage across the channel's α -subunit depolarizes and the β -subunit discharges NADP⁺ (right).

Disagreement between the voltage-clamp and current-clamp data

The authors did not see effects of the 4-ONE on the action potential firing pattern (Figure 5) although they showed that the inactivation time constant is augmented (Figure 3 and 4). They offered an explanation stating that it was more a technical issue rather than 4-ONE altering the firing pattern independently of Kv β (Page 8 last paragraph). This is a critical issue and must be addressed properly. Therefore, it raises the question: if such a technical issue is presented in the current-clamp experiments, why do the authors think that this is not an issue with the voltage-clamp measurements? Since they are patching the same cells, a technical issue would be present in both measurements and therefore jeopardizing their voltage-clamp/current-clamp data interpretation. This point raises the question whether the effects observed are more due to the ROS effects on PUFA rather than fragments generated by lipid oxidation (4-ONE, for example). This could be addressed by an experiment where one can use AOX and 4-ONE in the recording pipette, therefore capping ROS and making 4-ONE the only source of lipid peroxidation carbonyls as the substrate for the hyperkinetic enzymatic activity. In this context, why did the author focus exclusively on the enzymatic activity of hyperkinetic/shaker and did not consider the effects of PUFA on other voltage-gated ion channels? (Front Physiol. 2017 Feb 6;8:43.). Since PUFA modulates the activities of many ion channels, including K2Ps, the degradation of the PUFA would affect the function of ion channels expressed in the dFB neurons, including Sandman and Shaker. Thus, the authors must experimentally address the disagreements between the voltage-clamp and current-clamp data.

The revised Fig. 5c shows an anatomical image of dFBNs (yellow), superimposed on the distribution of endogenous Hk (turquoise), which carries an epitope tag after CRISPR–Cas9-mediated editing of the genomic locus. dFBN somata lie far laterally and are connected through long, thin processes first to dendrites, which extend dorsally, and then to axons, whose projections form an inverted V-shape in layers 6 and 7 of a midline structure, the dorsal fan-shaped body. A clearly visible pool of Hk colocalizes with these projections. Because 4-ONE readily reacts with endogenous nucleophiles, as indicated by our latest SMALDI-MSI experiments (please see Extended Data Fig. 7 and our response to Referee 1) and a reported half-life in tissue of <4 seconds^{3,4}, it is extremely unlikely that sufficient amounts of 4-ONE reach the distant axonal Hk population by diffusion from a somatically placed patch pipette. As a consequence, the cofactor of channels in the spike generation zone cannot be oxidized to NADP⁺, and the spiking response of dFBNs in current-clamp recordings reflects the continued presence of NADPH (Fig. 5b).

Current-clamp experiments after the illumination of miniSOG do not suffer this problem because the genetically encoded photosensitizer localizes throughout the neuronal arbor. Switching on the photogeneration of singlet oxygen results in an effective conversion of the axonal Hk population to the NADP⁺ form and a steepening of the input-output function, as seen before¹¹ (Fig. 5a).

In voltage-clamp experiments, axonal Sh channels far from the recording site contribute little to the measured current because the highly non-spherical geometry of dFBNs (Fig.

5c) interferes with the voltage activation of these channels²⁰. Voltage-clamp data are thus dominated by somatic channels, which are close enough to the patch pipette to allow cofactor oxidation by 4-ONE as well as by miniSOG (Fig. 3a, b). As far as we can see, there is no disagreement between current- and voltage-clamp data.

Our experiments do address the question ‘whether the effects observed are more due to the ROS effects on PUFA rather than fragments generated by lipid oxidation (4-ONE, for example).’ The illumination of miniSOG generates ROS, which initially attack the PUFA chains of membrane lipids; these peroxidized (but still backbone-linked) PUFAs then undergo a series of scissions and rearrangements that result in the release of carbonyls such as 4-ONE. The direct provision of 4-ONE bypasses this sequence of events and, therefore, the fragmentation of membrane lipids. The observation that the effects of miniSOG on the inactivation kinetics of I_A are indistinguishable from those of 4-ONE argues that ROS act via the production of carbonyls like 4-ONE and not via changes to the membrane itself.

Free PUFAs indeed regulate certain ion channels, as the reviewer notes, but there is no release or addition of free PUFAs in any of our experiments.

Access resistance (or series resistance) should be provided.

The authors stated in the methods section that, when the access resistance was larger than 50 M Ω (First paragraph page 17) the cells were discarded, and the pipette resistance was between 9-11M Ω (last sentence on page 16). Therefore, the access resistance in the experiments performed was between 10 and 50 M Ω . Under voltage-clamp, the membrane capacitance (Cm) and resistance (Rm) are in parallel, and both are in series with the access resistance (Ra). Based on Figures 3, 4, 6 and Extended figure 3, the average I_A current is around 500pA. Therefore, this would impose a voltage drop ($V_A = I_A * R_a$) due to Ra, ranging between 5mV (best case-scenario – Ra = 10M Ω) and 25mV (worst case scenario – 50 M Ω). The voltage drop due to the access resistance will provide serious discrepancies between the imposed voltage by the voltage command and the actual voltage applied in the cell membrane. For example, assuming the worst-case scenario, the voltage drop due to access resistance is 25mV and the voltage step used was from -110mV to +30mV (extended figure 2). The voltage that will be sensed in the membrane will be +5mV, instead of the +30mV imposed by the voltage command. This will also change the time course of the current because the error will change with the magnitude of the current over time. In this context, the authors must state whether they used a series resistance compensation and the amount used. More importantly, they should show a plot of the variability of the access resistance in the voltage-clamp experiments and discuss if these values could jeopardize their measurements.

The new Extended Data Fig. 4a plots series resistances and other biophysical parameters during 30-minute voltage-clamp recordings from dFBNs. These new recordings were performed in the presence of 50 μ M 4-ONE and included a series of depolarization steps between the 10- and 30-minute time points.

*We have revised the Methods section to state that series resistances were monitored but not compensated and allowed to rise by at most 20% (but never above 50 M Ω) during the course of a recording. Because series resistance compensation in whole-cell voltage-clamp recordings from central neurons in *Drosophila* is prone to instability, oscillations, and increased noise, it is standard practice to avoid it. However, we have added a cautionary remark to Methods about the voltage errors this is expected to cause.*

Exemplary current trace in the presence of 4-ONE and sogMini experiments. Although the authors have provided detailed statistical analysis of the currents recorded, the authors should also include in the figure 3, 4 and 6 exemplary current traces showing IA before and after the several treatments therein.

Examples of current traces are included in the revised Fig. 3, Fig. 6, and Extended Data Fig. 5, which together also cover the experimental conditions depicted in Fig. 4.

Minor concerns:

The figures regarding the voltage-clamp experiments are not easy to follow. The bars indicating that the cells were held at -80mV on the top panels of Figure 3, 4, 6 and extended data 3 is confusing. It is not clear what was the test pulse used to elicit the current and assess the membrane resistance, current amplitude and time constants. This gives the impression that there was no voltage change during the recording, which might mislead the readers.

We have attempted but failed to find an effective way to include the application of test pulses in the bars symbolizing the voltage commands during the recording because the time scales are so different. Instead, the revised legends to these figures mention that the voltage protocols depicted in Extended Data Fig. 2 were run at the times when currents were measured.

Shaker is a member of the Kv1 superfamily, but it is not the Kv1. The authors should rephrase the sentence when Shaker is introduced. Page 2: “A hint as a possible answer has come from studies in, where both the Kv1 channel Shaker and its β -subunit Hyperkinetic”.

We have corrected this sentence.

From the figures presented in 3, 4 and 6, it is not obvious that the authors are only recording IA. They should state in the text and also refer to Extended Figure 2 where they showed the procedure to isolate the IA currents from the other ionic components.

The revised legends to these figures mention that the voltage protocols depicted in Extended Data Fig. 2 were run at the indicated measurement times.

Throughout the voltage-clamp experiment the authors held the cell membrane at -80mV. Since, holding at -80mV in presence of 4-ONE or ROS allows the Kv β to load with NADP⁺ do the authors think that if membrane were held at +30mV would impede the Kv β to load with NADP⁺?

Experiments on rat Kv1.1 in complex with Kv β 1 suggest that NADPH oxidation is ~2-fold faster at a membrane potential of 0 mV than at -100 mV¹⁶.

References

1. Long, E. K., Olson, D. M. & Bernlohr, D. A. High-fat diet induces changes in adipose tissue trans-4-oxo-2-nonenal and trans-4-hydroxy-2-nonenal levels in a depot-specific manner. *Free Radic Biol Med* **63**, 390-398 (2013).

2. Harkin, C. et al. Spatial localization of β -unsaturated aldehyde markers in murine diabetic kidney tissue by mass spectrometry imaging. *Anal Bioanal Chem* **414**, 6657-6670 (2022).
3. Esterbauer, H., Schaur, R. J. & Zollner, H. Chemistry and biochemistry of 4-hydroxynonenal, malonaldehyde and related aldehydes. *Free Radic Biol Med* **11**, 81-128 (1991).
4. Parvez, S., Long, M. J. C., Poganik, J. R. & Aye, Y. Redox signaling by reactive electrophiles and oxidants. *Chem Rev* **118**, 8798-8888 (2018).
5. Sarnataro, R., Velasco, C. D., Monaco, N., Kempf, A. & Miesenböck, G. Mitochondrial origins of the pressure to sleep. *Submitted*. (2024).
6. Donlea, J. M., Pimentel, D. & Miesenböck, G. Neuronal machinery of sleep homeostasis in *Drosophila*. *Neuron* **81**, 860-872 (2014).
7. Pimentel, D. et al. Operation of a homeostatic sleep switch. *Nature* **536**, 333-337 (2016).
8. Ruan, H. & Wu, C. F. Social interaction-mediated lifespan extension of *Drosophila* Cu/Zn superoxide dismutase mutants. *Proc Natl Acad Sci U S A* **105**, 7506-7510 (2008).
9. DiChiara, T. J. & Reinhart, P. H. Redox modulation of hsl α Ca²⁺-activated K⁺ channels. *J Neurosci* **17**, 4942-4955 (1997).
10. Bushey, D., Huber, R., Tononi, G. & Cirelli, C. *Drosophila Hyperkinetic* mutants have reduced sleep and impaired memory. *J Neurosci* **27**, 5384-5393 (2007).
11. Kempf, A., Song, S. M., Talbot, C. B. & Miesenböck, G. A potassium channel β -subunit couples mitochondrial electron transport to sleep. *Nature* **568**, 230-234 (2019).
12. Chintaluri, C. & Vogels, T. P. Metabolically regulated spiking could serve neuronal energy homeostasis and protect from reactive oxygen species. *Proc Natl Acad Sci U S A* **120**, e2306525120 (2023).
13. Dennard, R. H. Field-effect transistor memory. US patent 3387286 (1968).
14. Dennard, R. H. How we made DRAM. *Nat Electron* **1**, 372 (2018).
15. Gulbis, J. M., Mann, S. & MacKinnon, R. Structure of a voltage-dependent K⁺ channel beta subunit. *Cell* **97**, 943-952 (1999).
16. Pan, Y., Weng, J., Cao, Y., Bhosle, R. C. & Zhou, M. Functional coupling between the Kv1.1 channel and aldoketoreductase Kv β 1. *J Biol Chem* **283**, 8634-8642 (2008).
17. Balaban, R. S., Nemoto, S. & Finkel, T. Mitochondria, oxidants, and aging. *Cell* **120**, 483-495 (2005).
18. Murphy, M. P. How mitochondria produce reactive oxygen species. *Biochem J* **417**, 1-13 (2009).

19. Chance, B., Sies, H. & Boveris, A. Hydroperoxide metabolism in mammalian organs. *Physiol Rev* **59**, 527-605 (1979).
20. Bar-Yehuda, D. & Korngreen, A. Space-clamp problems when voltage clamping neurons expressing voltage-gated conductances. *J Neurophysiol* **99**, 1127-1136 (2008).

Response to Reviewers of Nature Manuscript 2023-04-05806A

We thank the referees for their comments and suggestions. Changes in the revised manuscript are tracked in red.

Referee 1

The authors have fully addressed my issues around the lipidomic data and the 4-ONE analysis.

Referee 2

The authors have made genuine efforts to revise the manuscript in response to the reviewers' comments, and the manuscript has been greatly improved. There is still one major issue remaining that can be resolved straightforwardly with data collected from a few additional cell types.

The authors state that “Our model of homeostatic sleep regulation neither states nor demands that Sh-Hk is the only determinant of dFBN activity, nor that the mechanism of excitability control via Hk is exclusive to dFBNs. Our experiments concentrate on a particular ion channel in a specific type of neuron in order to test a sharply stated hypothesis: that Hk forms a voltage-gated lipid peroxidation memory whose contents in dFBNs influence sleep.” They resisted the suggestion to investigate the same physiological responses in non-dFBN neurons.

As the authors suggest, the Sh-Hk complex used as a lipid peroxidation sensor to convert the oxidative stress levels to neuronal excitability could be presumed as a common mechanism shared by many neurons. However, there is no guarantee that no additional factors or modified molecular properties specific to the dFBNs are involved to facilitate or tune the Sh-Hk complex sensor properties. There is no shortage of neuronal types in the fly brain that can be analyzed with the same physiologically experiments reported here. In fact, there are larger neurons that can be used for quick examinations.

It is indeed true that, as the authors pointed out in the analogy, all neurons use the common currency of action potentials to perform endless types of tasks. However, the essence is that different wave forms and firing patterns characteristic of different neuronal types are seen in the nervous system. To gain a deeper understanding of nervous system function, the action potential wave form and firing patterns should not be ignored.

Whereas the reader could agree with the proposal that the Sh-Hk complex is involved in the process, the natural question is whether it works the same way in dFBNs as in other neurons which all have the Sh-Hk complex in excitability control. To make this paper more interesting to a broader readership of the journal, the small investment in collecting data from some other neuronal types to resolve this issue could enhance the impact of this study.

To address this request, we initially sampled several non-dFB neuron types in the fly brain (olfactory projection neurons, mushroom body output neurons, and neurons of the pars intercerebralis [PI]) to determine i) whether the cells had sufficiently large and well-isolated A-type currents for measuring the impact of lipid peroxidation products on the inactivation kinetics of the channel and ii) whether they could tolerate 30- to 40-minute long voltage-clamp recordings that included 20 minutes of step depolarizations to

ascertain reversibility. Of the neuron types sampled, we found that only neurons of the PI met these criteria reasonably well (even though the slow inactivation time constant of I_A could not be measured accurately, due to the presence of a slowly activating potassium current, Extended Data Fig. 6a).

The new Extended Data Fig. 6a–e presents recordings from these cells. The inclusion of 4-ONE in the patch pipette elevated the fast inactivation time constant, as it does in dFBNs; a simulated 20-minute spike train returned the time constant to baseline, again recapitulating the situation in dFBNs. To observe this voltage-dependent reversal of channel inactivation kinetics in PI neurons, we found it necessary to reduce the 4-ONE concentration in the intracellular solution from 50 μ M to 1 μ M. Because PI neurons are significantly larger than dFBNs, patch electrodes with lower resistances were used. Rapid dialysis of the cytoplasm led to quick re-oxidation of the cofactor bound to the Sh-Hk complex at the higher 4-ONE concentration.

The demonstration that the same carbonyl- and voltage-driven changes in inactivation kinetics are present in HEK-293 cells expressing mouse $K_V1.4$ and $K_V\beta2$ (Extended Data Fig. 6f–j) provides the strongest conceivable evidence that no additional factors or modified molecular properties specific to dFBNs are needed to endow a Shaker-family K_V1 channel with the ability to sense lipid-derived carbonyls.

Referee 4

The authors have significantly improved the text and the revised version was easier to follow, but there are still places that need further clarification. I recommend the incorporation of figure B (shown in the rebuttal) since it depicts in a simple manner the DRAM analogy and the proposed mechanism.

We have included the schematic illustration of the DRAM analogy and proposed mechanism as the new Fig. 1.

There is no question about the quality of the work and the care that the researchers took in conducting the experiments. While the manuscript presents compelling evidence supporting their hypothesis, the discrepancies between the current-clamp and voltage-clamp data undermine the robustness of the conclusions drawn. The voltage-clamp data is extensively tested and using different controls, which is lacking in the current-clamp data. In my opinion, I respectfully argue that the current-clamp data is the most important evidence for their hypothesis since it reflects more closely the physiological condition and response of the dFBNs. I do understand the experimental challenges that the authors are dealing with when performing electrophysiological assays in the dFBNs. The space-clamp issue that they reported as one of the possible explanations for the lack of effects in the 4-ONE experiments will be present in both current and voltage-clamp experiments. While it is appreciated that the authors acknowledged difficulties and tried to provide explanation for their technical issues, I disagree with their explanation. Thus, I recommend that they address the following concerns regarding their explanation for the current-clamp data which, in my opinion, was not addressed by the authors in the revised form of the manuscript and will increase the robustness of their conclusions.

To settle the question of why the inclusion of synthetic 4-ONE in the patch pipette alters the inactivation kinetics of the A-type current but fails to increase the spike rate of dFBNs

during injections of depolarizing current, we have obtained voltage- and current-clamp recordings from dFBNs in sniffer¹ mutant males and wild-type controls. Because 4-ONE is an endogenous substrate of the carbonyl reductase encoded by the sniffer gene, its intracellular concentration increases as a consequence of the enzyme's absence. In contrast to exogenous 4-ONE, which must reach its ion channel targets by diffusion from a patch pipette placed at the soma, the local release of endogenous lipid peroxidation products throughout the neuronal arbor is not expected to face this barrier. We show in new voltage-clamp and current-clamp recordings that this is indeed the case: carriers of the mutant *sni*¹ allele have elevated fast and slow I_A inactivation time constants (Fig. 4a, b) that go hand-in-hand with increases in the excitability of dFBNs (Fig. 6a) and increases in sleep (Fig. 3).

1) The current-clamp and voltage-clamp disagreement data still needs clarification. The fact the 4-ONE is still active in the recording for the voltage clamp experiments even after 40 minutes of the experiments (Fig. 6d), does not align with the idea of half-life in tissue (<4s) as the cause of the lack of effect of 4-ONE in the spike response shown current-clamp experiments (Fig 5b). Thus, I respectfully disagree about the short half-life of the 4-ONE as the reason for the current-clamp data technical issue postulated in the page 9.

Extended Data Fig. 7 shows that 4-ONE is stable for hours in contact with glass but vanishes without a trace after contact with brain tissue, due to reaction with endogenous nucleophiles, enzymatic conversion, or both. In our patch-clamp recordings, the glass electrode contains a large reservoir of 4-ONE, which remains active for the duration of the experiment. 4-ONE molecules diffusing from the glass pipette into the recorded neuron can therefore exert the same effect on nearby channels after 40 minutes as they do at the beginning of the experiment. However, once inside dFBNs 4-ONE is unable to modify ion channels at some distance from the soma because the diffusion times needed to reach these locations exceed its half-life in the cell.

2) The authors also justify the lack of increase in the spike response due to the low capability of the diffusion of the 4-ONE to the dendrites and axons, which does not happen in the miniSOG scenario. Thus, one can conclude that the dendrites and axons can be more effective in spike initiation in the dFBNs than the soma does. In order to demonstrate the lack of the effects of the 4-ONE due to the poor diffusion to the dendrites and axons, the authors can perform control experiments where they use miniSOG. Since miniSOG would be present in the soma, dendrites, and axons they could illuminate different regions to support their interpretation regarding the poor diffusion of 4-ONE. 1) They can illuminate the dendrites and axons regions only. This will test the hypothesis that those regions are more effective than the soma to increase the spike response. 2) They can only illuminate the soma when miniSOG is present. In this condition only Shaker present at the soma will be affected by ROS, and according to their explanation for poor diffusion of 4-ONE, this will not lead to an increase in the spike response. 3) They must run a control where they illuminate the dFBNs only with LED without the expression of miniSOG to fully demonstrate that the LED by itself is not leading to the increase in the spike response by unintended effects such as increase in temperature, for example. This needs to be done using the same illumination condition that they tested in the data reported in this study.

dFBNs possess the characteristic morphology and physiology of central neurons in Drosophila: a thin primary neurite emanating from the soma divides into dendritic and

axonal branches (Fig. 6d). The spike initiation zone typically lies at the beginning of the axonal branch (Gouwens and Wilson, *J. Neurosci.* 29: 6239, 2009), which in the case of dFBNs is separated from the soma by $\sim 100 \mu\text{m}$ (Fig. 6d). The slow, attenuated action potentials recorded at the soma (Fig. 6a–c) reflect this electrotonic distance.

An experiment in which only specific subcellular compartments of miniSOG-expressing dFBNs, such as axons or somata, are illuminated is technically impossible because LED light cannot be patterned at such fine spatial resolution in scattering tissue.

As an alternative strategy to test whether exogenous 4-ONE delivered through the patch pipette fails to reach the spike initiation zone, we have used the sniffer¹ mutation to elevate levels of endogenous 4-ONE throughout the neuronal arbor and observed elevated spike rates relative to wild-type controls (Fig. 6a).

The delivery of 4-ONE through the patch pipette is the only experimental intervention we have encountered in which changes in channel inactivation are dissociated from changes in spike rate, creating an ostensible disagreement. The sniffer¹ mutation in the current manuscript (Fig. 4a, b and Fig. 6a) as well as several manipulations in Fig. 4 of our earlier study (Kempf et al., *Nature* 2019) all produce coordinated effects on dFBN spike frequency and potassium channel inactivation: illumination of miniSOG, the expression of pro- or antioxidant enzymes, and the elimination or restoration of redox-sensitivity from the Shaker channel. Common to these manipulations is that they target endogenous proteins or rely on genetically encoded tools, which can exert effects in all parts of the cell. Diffusion of 4-ONE from a point source at the soma fundamentally can not, and there is no experimental remedy for this limitation.

The “control where they illuminate the dFBNs only with LED without the expression of miniSOG to fully demonstrate that the LED by itself is not leading to the increase in the spike response” was performed exactly as requested as part of the first use of miniSOG in dFBNs (Kempf et al., *Nature* 2019, Extended Data Fig. 4).

Figure 5 c-e: it is hard to understand what the authors are trying to show in the figure. Are they trying to compare different expression levels of Hyperkinetic or just to show how the soma, dendrites and axons are distributed? To improve clarity, they can label in the figure the soma, the dendrites, and axons so the paper can be more appreciated by readers not familiar with the anatomy of the drosophila brain.

The figure (now Fig. 6d–f) shows that the expression of Hyperkinetic in the brain is widespread but notably high in the fan-shaped body layer innervated by dFBNs, and it introduces non-expert readers to the anatomy of these neurons. We appreciate the suggestion of labelling the different neuronal compartments, which we have adopted.

Minor comments:

Extended Figure 6: what are the conditions of the traces (gray and the turquoise, and yellow and turquoise, respectively) in b and d? Do they see the same rundown effects as in the IA? It is appreciated that the authors did controls using the mouse Kv1.4 and Kv β 2 subunit, however, I wonder why the Shaker and the Hyperkinetic for control weren't used, since they would be the ideal system in this case.

We have modified Extended Data Fig. 6g, i and its legend to make clear that the turquoise traces show examples of peak-normalized currents after methylglyoxal exposure, whereas

the gray and yellow traces represent peak-normalized currents in the same cells 20 minutes later, in the absence (gray, panel g) or presence (yellow, panel i) of a series of depolarization steps. The amplitude of I_A runs down in the same manner in HEK293 cells as it does in dFBNs (Fig. 6h, j).

The reason for studying the mouse $K_V1.4$ – $K_V\beta 2$ complex in HEK293 was to demonstrate that mammalian K_V1 channels also form voltage-gated lipid peroxidation memories.

Response to Reviewers of Nature Manuscript 2023-04-05806B

We thank the referees for their comments.

Referee 2

In this revised manuscript, the authors have satisfactorily addressed the issues raised in my previous review.

Referee 4

The authors have addressed my concerns. They added new experimental data in the manuscript that strength their hypothesis. I fully support the publication.